# Boosting Multiagent Reinforcement Learning via Permutation Invariant and Permutation Equivariant Networks

**Jianye Hao**[1][2]**, Xiaotian Hao**[1]**, Hangyu Mao**[2]**, Weixun Wang**[1]**, Yaodong Yang**[3]**, Dong Li**[2]**,
Yan Zheng**[1]**, Zhen Wang**[4]
[1]College of Intelligence and Computing, Tianjin University, [2]Noah's Ark Lab, Huawei
[3]The Chinese University of Hong Kong, [4]Northwestern Polytechnical University

## Abstract

The state space in Multiagent Reinforcement Learning (MARL) grows exponentially with the agent number. Such a curse of dimensionality results in poor scalability and low sample efficiency, inhibiting MARL for decades. To break this curse, we propose a unified agent permutation framework that exploits the permutation invariance (PI) and permutation equivariance (PE) inductive biases to reduce the multiagent state space. Our insight is that permuting the order of entities in the factored multiagent state space does not change the information. Specifically, we propose two novel implementations: a Dynamic Permutation Network (DPN) and a Hyper Policy Network (HPN). The core idea is to build separate entity-wise PI input and PE output network modules to connect the entity-factored state space and action space in an end-to-end way. DPN achieves such connections by two separate module selection networks, which consistently assign the same input module to the same input entity (guarantee PI) and assign the same output module to the same entity-related output (guarantee PE). To enhance the representation capability, HPN replaces the module selection networks of DPN with hypernetworks to directly generate the corresponding module weights. Extensive experiments in SMAC, SMACv2, Google Research Football, and MPE validate that the proposed methods significantly boost the performance and the learning efficiency of existing MARL algorithms. Remarkably, in SMAC, we achieve **100%** win rates in almost all hard and super-hard scenarios (never achieved before).

## 1 Introduction

Multiagent Reinforcement Learning (MARL) has successfully addressed many real-world problems (Vinyals et al., 2019; Berner et al., 2019; Hüttenrauch et al., 2017). However, MARL algorithms still suffer from poor sample-efficiency and poor scalability due to the curse of dimensionality, i.e., the joint state-action space grows exponentially as the agent number increases (Li et al., 2022). A way to solve this problem is to properly reduce the size of the state-action space (van der Pol et al., 2021; Li et al., 2021). In this paper, we study how to utilize the permutation invariance (PI) and permutation equivariance (PE)[1] inductive biases to reduce the state space in MARL.

Let $G$ be the set of all permutation matrices[2] of size $m \times m$ and $g$ be a specific permutation matrix of $G$. A function $f : X \to Y$ where $X = [x_1, \ldots x_m]^\mathsf{T}$, is PI if permutation of the input components does not change the function output, i.e., $f(g[x_1, \ldots x_m]^\mathsf{T}) = f([x_1, \ldots x_m]^\mathsf{T}), \forall g \in G$. In contrast, a function $f : X \to Y$ where $X = [x_1, \ldots x_m]^\mathsf{T}$ and $Y = [y_1, \ldots y_m]^\mathsf{T}$, is PE if permutation of the input components also permutes the outputs with the same permutation $g$, i.e., $f(g[x_1, \ldots x_m]^\mathsf{T}) = g[y_1, \ldots y_m]^\mathsf{T}, \forall g \in G$. For functions that are not PI or PE, we uniformly denote them as permutation-sensitive functions.

---

[1]For brevity, we use PI/PE as abbreviation of permutation invariance/permutation equivariance (nouns) or permutation-invariant/permutation-equivariant (adjectives) depending on the context.

[2]A permutation matrix has exactly a single unit value in every row and column and zeros everywhere else.

A multiagent environment typically consists of $m$ individual entities, including $n$ learning agents and $m - n$ non-player objects. The observation $o_i$ of each agent $i$ is usually composed of the features of the $m$ entities, i.e., $[x_1, \ldots x_m]$, where $x_i \in \mathcal{X}$ represents each entity's features and $\mathcal{X}$ is the feature space. If simply representing $o_i$ as a concatenation of $[x_1, \ldots x_m]$ in a fixed order, the observation space will be $|\mathcal{X}|^m$. A prior knowledge is that although there are $m!$ different orders of these entities, they inherently have the same information. Thus building functions that are insensitive to the entities' orders can significantly reduce the observation space by a factor of $\frac{1}{m!}$. To this end, in this paper, we exploit both PI and PE functions to design more sample efficient MARL algorithms.

Take the individual Q-network of the StarCraft Multiagent Challenge (SMAC) benchmark (Samvelyan et al., 2019) $Q_i(a_i|o_i)$ as an example. As shown in Fig.1, the input $o_i$ consists of two groups of entities: an ally group $o_i^{\text{ally}}$ and an enemy group $o_i^{\text{enemy}}$. The outputs consist of two groups as well, i.e., Q-values for move actions $\mathcal{A}_i^{\text{move}}$ and Q-values for attack actions $\mathcal{A}_i^{\text{attack}}$. Given the same $o_i$ arranged in different orders, the Q-values of $\mathcal{A}_i^{\text{move}}$ should be kept the same, thus we can use PI architectures to make $Q_i(\mathcal{A}_i^{\text{move}}|o_i)$ learn more efficiently. For $Q_i(\mathcal{A}_i^{\text{attack}}|o_i)$, since there exists a one-to-one correspondence between each enemy's features in $o_i^{\text{enemy}}$ and each attack action in $\mathcal{A}_i^{\text{attack}}$, permutations of $o_i^{\text{enemy}}$ should result in the same permutations of $\mathcal{A}_i^{\text{attack}}$, so we can use PE architectures to make $Q_i(\mathcal{A}_i^{\text{attack}}|o_i)$ learn more efficiently.

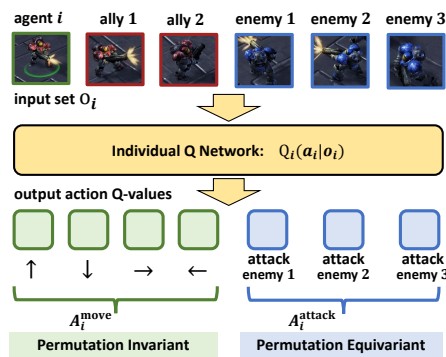

Figure 1: A motivation example in SMAC.

To achieve PI, there are two types of previous methods. The first employs the idea of data augmentation, e.g., Ye et al. (2020) propose data augmented MADDPG, which generates more training data by shuffling the order of the input components and forcedly maps these generated data to the same output through training. But it is inefficient to train a permutation-sensitive function to output the same value when taking features in different orders as inputs. The second type applies naturally PI architectures, such as Deep Sets (Li et al., 2021) and GNNs (Wang et al., 2020b; Liu et al., 2020), to MARL. These models use shared input embedding layers and entity-wise pooling layers to achieve PI. However, using shared embedding layers limits the model's representational capacity and may result in poor performance (Wagstaff et al., 2019). For PE, to the best of our knowledge, it has drawn relatively less attention in MARL community and few works exploit this property.

In general, the architecture of an agent's policy network can be considered as three parts: ❶ an input layer, ❷ a backbone network (main architecture) and ❸ an output layer. To achieve PI and PE, we follow the *minimal modification principle* and propose a light-yet-efficient agent permutation framework, where we only modify the input and output layers while keeping backbone unchanged. Thus our method can be more easily plugged into existing MARL methods. The core idea is that, instead of using shared embedding layers, we build non-shared entity-wise PI input and PE output network modules to connect the entity-factored state space and action space in an end-to-end way.

Specifically, we propose two novel implementations: a Dynamic Permutation Network (DPN) and a Hyper Policy Network (HPN). To achieve PI, DPN builds a separate module selection network, which consistently selects the same input module for the same input entity no matter where the entity is arranged and then merges all input modules' outputs by sum pooling. Similarly, to achieve PE, it builds a second module selection network, which always assigns the same output module to the same entity-related output. However, one restriction of DPN is that the number of network modules is limited. As a result, the module assigned to each entity may not be the best fit. To relax the restriction and enhance the representational capability, we further propose HPN which replaces the module selection networks of DPN with hypernetworks and directly generates the network parameters of the corresponding modules (by taking each entity's own features as input). Entities with different features are processed by modules with entity-specific parameters. Therefore, the model's representational capability is improved while ensuring the PI and PE properties.

Extensive evaluations in SMAC, SMACv2, Google Research Football and MPE validate that DPN and HPN can be easily integrated into many existing MARL algorithms (both value-based and policy-based) and significantly boost their learning efficiency and converged performance. Remarkably, we achieve **100%** win-rates in almost all hard and super-hard scenarios of SMAC,

which has **never been achieved before** to the best of our knowledge. The code is available at https://github.com/tjuHaoXiaotian/API-Network.

## 2 RELATED WORK

To highlight our method, we briefly summarize the related works that consider the PI or PE property.

**Concatenation**. Typical MARL algorithms, e.g., QMIX (Rashid et al., 2018) and MADDPG (Lowe et al., 2017) simply represent the set input as a concatenation of the $m$ entities' features in a fixed order and feed the concatenated features into permutation-sensitive functions, e.g., multilayer perceptron (MLP). As each entity's feature space size is $|\mathcal{X}|$, the size of the joint feature space after concatenating will grow exponentially to $|\mathcal{X}|^m$, thus these methods suffer sample inefficiency.

**Data Augmentation**. To reduce the number of environmental interactions, Ye et al. (2020) propose data augmented MADDPG which generates more training data by shuffling the order of $[x_1, \ldots x_m]$ and additionally updates the model based on the generated data. However, the method requires more computational resources and is more time-consuming. Besides, as the generated data contains the same information as the original one, they should have the same Q-value. But it is inefficient to train a permutation-sensitive function to output the same value when taking differently-ordered inputs.

**Deep Set & Graph Neural Network**. Instead of doing data augmentation, Deep Set (Zaheer et al., 2017) constructs a family of PI neural architectures for learning set representations. Each component $x_i$ is mapped separately to some latent space using a shared embedding layer $\phi(x_i)$. These latent representations are then merged by a PI pooling layer (e.g. sum, mean) to ensure the PI of the whole function, e.g., $f(X) = \rho\left(\Sigma_{i=1}^m \phi(x_i)\right)$, where $\rho$ can be any function. Graph Neural Networks (GNNs) (Veličković et al., 2018; Battaglia et al., 2018) also adopt shared embedding and pooling layers to learn functions on graphs. Wang et al. (2020b); Li et al. (2021) and Jiang et al. (2018); Liu et al. (2020) have applied Deep Set and GNN to MARL. However, due to the use of the shared embedding $\phi(x_i)$, the representation capacity is usually limited (Wagstaff et al., 2019).

**Multi-head Self-Attention & Transformer.** To improve the representational capacity, Set Transformer (Lee et al., 2019) employs the multi-head self-attention mechanism (Vaswani et al., 2017) to process every $x_i$ of the input set, which allows the method to encode higher-order interactions between elements in the set. Most recently, Hu et al. (2021b) adopt Transformer (Vaswani et al., 2017) to MARL and proposes UPDeT, which could handle various input sizes. But UPDeT is originally designed for transfer learning scenarios and does not explicitly consider the PI and PE properties.

**PE Functions in MARL.** In the deep learning literature, some works have studied the effectiveness of PE functions when dealing with problems defined over graphs (Maron et al., 2018; Keriven & Peyré, 2019). However, in MARL, few works exploit the PE property to the best of our knowledge. One related work is Action Semantics Network (ASN) (Wang et al., 2019), which studies the different effects of different types of actions but does not directly consider the PE property.

## 3 PROBLEM STATEMENT

### 3.1 ENTITY-FACTORED MODELING IN DEC-POMDP

A cooperative multiagent environment typically consists of $m$ entities, including $n$ learning agents and $m-n$ non-player objects. We follow the definition of Dec-POMDP (Oliehoek & Amato, 2016). At each step, each agent $i$ receives an observation $o_i \in \mathcal{O}_i$ which contains partial information of the state $s \in \mathcal{S}$, and executes an action $a_i \in \mathcal{A}_i$ according to a policy $\pi_i(a_i|o_i)$. The environment transits to the next state $s'$ and all agents receive a shared global reward. The target is to find optimal policies for all agents which can maximize the expected cumulative global reward. Each agent's individual action-value function is denoted as $Q_i(o_i, a_i)$. Detailed definition can be found in Appendix B.1.

Modeling MARL in factored spaces is a common practice. Many recent works (Qin et al., 2022; Wang et al., 2020b; Hu et al., 2021b; Long et al., 2019; Wang et al., 2019) model the observation and the state of typical MARL benchmarks e.g., SMAC (Qin et al., 2022; Hu et al., 2021b), MPE (Long et al., 2019) and Neural MMO (Wang et al., 2019), into factored parts relating to the environment, agent itself and other entities. We follow such a common entity-factored setting that both the state space and the observation space are factorizable and can be represented as entity-related features,

i.e., $s \in \mathcal{S} \subseteq \mathbb{R}^{m \times d_s}$ and $o_i \in \mathcal{O} \subseteq \mathbb{R}^{m \times d_o}$, where $d_s$ and $d_o$ denote the feature dimension of each entity in the state and the observation. For the action space, we consider a general setting that each agent's actions are composed of two types: a set of $m$ entity-correlated actions $\mathcal{A}_{\text{equiv}}$[3] and a set of entity-uncorrelated actions $\mathcal{A}_{\text{inv}}$, i.e., $\mathcal{A}_i \triangleq (\mathcal{A}_{\text{equiv}}, \mathcal{A}_{\text{inv}})$. The entity-correlated actions mean that there exists a one-to-one correspondence between each entity and each action, e.g., 'attacking which enemy' in SMAC or 'passing the ball to which teammate' in football games. Therefore, $a_{\text{equiv}} \in \mathcal{A}_{\text{equiv}}$ should be equivariant with the permutation of $o_i$ and $a_{\text{inv}} \in \mathcal{A}_{\text{inv}}$ should be invariant. Tasks only considering one of the two types of actions can be considered special cases.

## 3.2 OUR TARGET: DESIGNING PI AND PE POLICY NETWORKS

Let $g \in G$ be an arbitrary permutation matrix. We define that $g$ operates on $o_i$ by permuting the orders of the $m$ entities' features, i.e., $go_i$, and that $g$ operates on $a_i = (a_{\text{equiv}}, a_{\text{inv}})$ by permuting the orders of $a_{\text{equiv}}$ but leaving $a_{\text{inv}}$ unchanged, i.e., $ga_i \triangleq (ga_{\text{equiv}}, a_{\text{inv}})$. Our target is to inject the PI and PE inductive biases into the policy network design such that:

$$\pi_i(a_i|go_i) = g\pi_i(a_i|o_i) \triangleq (g\pi_i(a_{\text{equiv}}|o_i), \pi_i(a_{\text{inv}}|o_i)) \qquad \forall g \in G, o_i \in \mathcal{O} \qquad (1)$$

where $g$ operates on $\pi_i$ by permuting the orders of $\pi_i(a_{\text{equiv}}|o_i)$ while leaving $\pi_i(a_{\text{inv}}|o_i)$ unchanged.

# 4 METHODOLOGY

## 4.1 THE HIGH-LEVEL IDEA

Our target policy network architecture is shown in Fig.2, which consists of four modules: ❶ input module $A$, ❷ backbone module $B$ which could be any architecture, ❸ output module $C$ for actions $a_{\text{inv}}$ and ❹ output module $D$ for actions $a_{\text{equiv}}$. For brevity, we denote the inputs and outputs of these modules as: $z_i = A(o_i), h_i = B(z_i), \pi_i(a_{\text{inv}}|o_i) = C(h_i)$, and $\pi_i(a_{\text{equiv}}|o_i) = D(h_i)$ respectively.

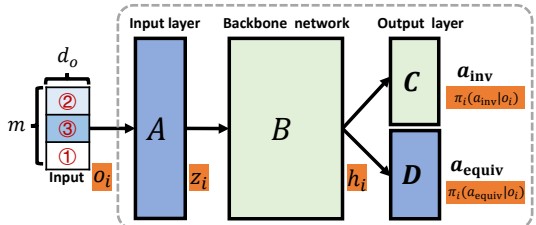

Figure 2: The ideal PI and PE policy network.

To achieve equation 1, we have to modify the architectures of $\{A, B, C, D\}$ such that the outputs of $C$ are PI and the outputs of $D$ are PE. In this paper, we propose to directly modify $A$ to be PI and modify $D$ to be PE with respect to the input $o_i$, and keep the backbone module B and output module C unchanged. The following Propositions show that our proposal is a feasible and simple solution.

**Proposition 1** *If we make module $A$ become PI, the output of module $C$ will immediately become PI without modifying module $B$ and $C$.*

*Proof.* Given two different inputs $go_i$ and $o_i, \forall g \in G$, since module $A$ is PI, then $A(go_i) = A(o_i)$. Accordingly, for any functions $B$ and $C$, we have $(C \circ B \circ A)(go_i) = (C \circ B \circ A)(o_i)$, which is exactly the definition of PI.

*Corollary 1.* With module $A$ being PI, if not modifying module $D$, the output of $D$ will also immediately become PI, i.e., $(D \circ B \circ A)(go_i) = (D \circ B \circ A)(o_i), \forall g \in G$.

**Proposition 2** *With module $A$ being PI, to make the output of module $D$ become PE, we must introduce $o_i$ as an additional input.*

*Proof.* According to corollary 1, with module $A$ being PI, module $D$ also becomes PI. To convert module $D$ into PE, we must have module $D$ know the entities' order in $o_i$. Thus we have to add $o_i$ as an additional input to $D$, i.e., $\pi_i(a_{\text{equiv}}|o_i) = D(h_i, o_i)$. Then, we only have to modify the architecture of $D$ such that $D(h_i, go_i) = gD(h_i, o_i) = g\pi_i(a_{\text{equiv}}|o_i), \forall g \in G$.

---

[3]Note that the size of $\mathcal{A}_{\text{equiv}}$ can be smaller than $m$, i.e., only corresponding to a subset of the entities (e.g., enemies or teammates). We use $m$ here for the target of simplifying notations.

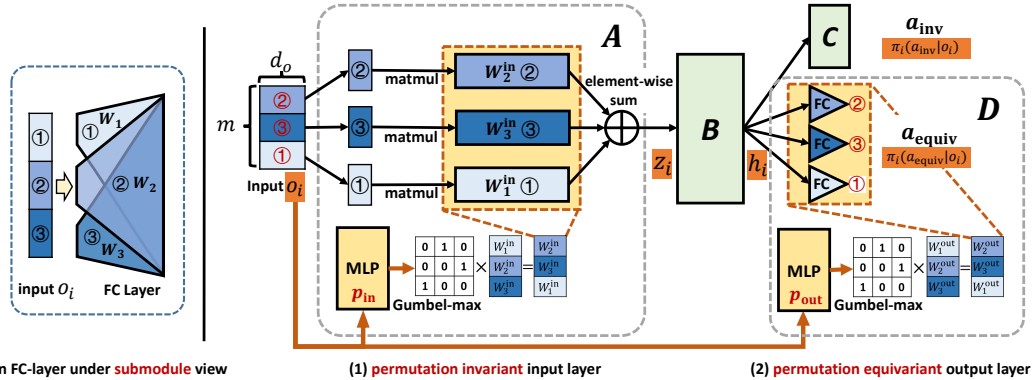

Figure 3: an FC-layer in submodule view (left); dynamic permutation network architecture (right).

**Minimal Modification Principle.** Although modifying different parts of $\{A, B, C, D\}$ may also achieve equation 1, the main advantage of our proposal is that we can keep the backbone module $B$ (and output module $C$) unchanged. Since existing algorithms have invested a lot in handling MARL-specific problems, e.g., they usually incorporate Recurrent Neural Networks (RNNs) into the backbone module $B$ to handle the partially observable inputs, we believe that achieving PI and PE without modifying the backbone architectures of the underlying MARL algorithms is beneficial. Following this minimal modification principle, our proposed method can be more easily plugged into many types of existing MARL algorithms, which will be shown in section 5. In the following, we propose two designs to implement the PI module $A$ and the PE module $D$.

## 4.2 DYNAMIC PERMUTATION NETWORK

As a common practice adopted by many MARL algorithms, e.g., QMIX (Rashid et al., 2018), MAD-DPG (Lowe et al., 2017) and MAPPO (Yu et al., 2021), module $A$, $C$ and $D$ are Fully Connected (FC) Layers and module $B$ is a Deep Neural Network (DNN) which usually incorporates RNNs to handle the partially observable inputs. Given an input $o_i$ containing $m$ entities' features, an FC-layer under a submodule view is shown in Fig.3 (left), which has $m$ independent weight matrices. If we denote $\mathcal{W}_{\text{in}} = \left[\mathbf{W}_1^{\text{in}}, \dots \mathbf{W}_m^{\text{in}}\right]^{\mathsf{T}}$ as the weights of the FC-layer $A$, the output is computed as[4]:

$$z_i = \sum_{j=1}^{m} o_i[j] \mathcal{W}_{\text{in}}[j] \tag{2}$$

where $[j]$ indicates the $j$-th element. Similarly, we denote $\mathcal{W}_{\text{out}} = \left[\mathbf{W}_1^{\text{out}}, \dots \mathbf{W}_m^{\text{out}}\right]^{\mathsf{T}}$ as the weight matrices of the output layer $D$. The output $\pi_i(a_{\text{equiv}}|o_i)$ is computed as:

$$\pi_i(a_{\text{equiv}}|o_i)[j] = h_i \mathcal{W}_{\text{out}}[j] \qquad \forall j \in \{1, \dots, m\} \tag{3}$$

**PI Input Layer $A$.** According to equation 2, permuting $o_i$, i.e., $o_i' = g o_i$, will result in a different output $z_i'$, as the j-th input is changed to $o_i'[j]$ while the j-th weight matrix remains unchanged. To make an FC-layer become PI, we design a PI weight matrix selection strategy for each $o_i[j]$ such that no matter where $o_i[j]$ is arranged, the same $o_i[j]$ will always be multiplied by the same weight matrix. Specifically, we build a weight selection network of which the output dimension is $m$:

$$p_{\text{in}}\left(\left[\mathbf{W}_1^{\text{in}}, \dots \mathbf{W}_m^{\text{in}}\right]\big|o_i[j]\right) = \text{softmax}(\text{MLP}\left(o_i[j]\right)) \tag{4}$$

where the $k$-th output $p_{\text{in}}\left(\mathbf{W}_k^{\text{in}}\big|o_i[j]\right)$ indicates the probability that $o_i[j]$ selects the $k$-th weight matrix of $\mathcal{W}_{\text{in}}$. Then, for each $o_i[j]$, we choose the weight matrix with the maximum probability. However, directly selecting the argmax index is not differentiable. To make the selection process trainable, we apply a Straight Through Estimator (Van Den Oord et al., 2017) to get the one-hot encoding of the argmax index. We denote it as $\hat{p}_{\text{in}}\left(o_i[j]\right)$. The weight matrix with the maximum probability can be acquired by $\hat{p}_{\text{in}}\left(o_i[j]\right) \mathcal{W}_{\text{in}}$, i.e. selecting the corresponding row from $\mathcal{W}_{\text{in}}$. Overall, no matter which orders the input $o_i$ is arranged, the output of layer A is computed as:

$$z_i = \sum_{j=1}^{m} o_i'[j] \; \left(\hat{p}_{\text{in}}\left(o_i'[j]\right) \mathcal{W}_{\text{in}}\right) \qquad o_i' = g o_i, \; \forall g \in G \tag{5}$$

---

[4]For brevity and clarity, all linear layers are described without an explicit bias term; adding one does not affect the analysis.

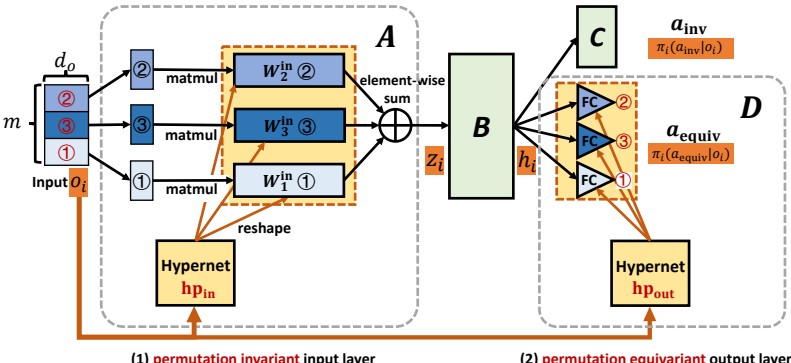

Figure 4: PI and PE network with hypernetworks.

Since the selection network only takes each entity's features $o_i[j]$ as input, the same $o_i[j]$ will always generate the same probability distribution and thus the same weight matrix will be selected no matter where $o_i[j]$ is ranked. Therefore, the resulting $z_i$ remains the same regardless of the arranged orders of $o_i$, i.e., layer A becomes PI. An illustration architecture is shown in Fig.3 (right).

**PE Output Layer $D$.** To make the output layer $D$ achieve PE, we also build a weight selection network $\hat{p}_{\text{out}}(o_i[j])$ for each entity-related output. The $j$-th output of $D$ is computed as:

$$\pi_i(a_{\text{equiv}}|o_i')[j] = h_i(\hat{p}_{\text{out}}(o_i'[j])\,\mathcal{W}_{\text{out}}) \qquad o_i' = go_i, \forall g \in G, \forall j \in \{1, \ldots, m\} \qquad (6)$$

For $\forall g \in G$, the $j$-th element of $o_i'$ will always correspond to the same matrix of $\mathcal{W}_{\text{out}}$ and thus the $j$-th output of $\pi_i(a_{\text{equiv}}|o_i')$ will always have the same value. The input order change will result in the same output order change, thus achieving PE. The architecture is shown in Fig.3 (right).

### 4.3 Hyper Policy Network

The core idea of DPN is to always assign the same weight matrix to each $o_i[j]$. Compared with Deep Set style methods which use a single shared weight matrix to embed the input $o_i$, i.e., $|\mathcal{W}_{\text{in}}| = 1$, DPN's representational capacity has been improved, i.e., $|\mathcal{W}_{\text{in}}| = m$. However, the sizes of $\mathcal{W}_{\text{in}}$ and $\mathcal{W}_{\text{out}}$ are still limited. One restriction is that for each $o_i[j]$, we can only select weight matrices from these limited parameter sets. Thus, the weight matrix assigned to each $o_i[j]$ may not be the best fit.

One question is whether we can provide an infinite number of candidate weight matrices such that the solution space of the assignment is no longer constrained. To achieve this, we propose a Hyper Policy Network (HPN), which incorporates hypernetworks (Ha et al., 2016) to generate customized embedding weights for each $o_i[j]$, by taking $o_i[j]$ as input. Hypernetworks (Ha et al., 2016) are a family of neural architectures which use one network to generate the weights for another network. Since we consider the outputs of hypernetworks as the weights in $\mathcal{W}_{\text{in}}$ and $\mathcal{W}_{\text{out}}$, the sizes of these parameter sets are no longer limited to $m$. To better understand our motivations, we provide a simple policy evaluation experiment in Appendix A.1, where we can directly construct an infinite number of candidate weight matrices, to show the influence of the model's representational capacity.

**PI Input Layer $A$.** For the input layer $A$, we build a hypernetwork $\text{hp}_{\text{in}}(o_i[j]) = \text{MLP}(o_i[j])$ to generate a corresponding $\mathcal{W}_{\text{in}}[j]$ for each $o_i[j]$.

$$z_i = \sum_{j=1}^{m} o_i'[j]\,\text{hp}_{\text{in}}(o_i'[j]) \qquad o_i' = go_i, \ \forall g \in G \qquad (7)$$

As shown in Fig.4, we first feed all $o_i[j]$s (colored by different blues), into the shared hypernetwork $\text{hp}_{\text{in}}(o_i[j])$ (colored in yellow), whose input size is $d_o$ and output size is $d_o d_h$[5]. Then, we reshape the output for each $o_i[j]$ to $d_o \times d_h$ and regard it as the weight matrix $\mathcal{W}_{\text{in}}[j]$. Different $o_i[j]$s will generate different weight matrices and the same $o_i[j]$ will always correspond to the same one no matter where it is arranged. The output of layer $A$ is computed according to equation 7. Since each $o_i[j]$ is embedded separately by its corresponding $\mathcal{W}_{\text{in}}[j]$ and then merged by a PI 'sum' function, the PI property is ensured.

---

[5] $d_o$ is the dimension of $o_i[j]$ and $d_h$ is the dimension of $h_i$.

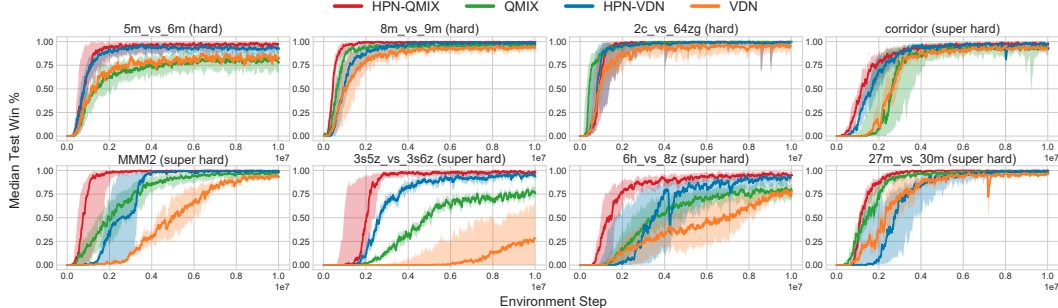

Figure 5: Comparisons of HPN-QMIX, HPN-VDN against fine-tuned QMIX and fine-tuned VDN.

**PE Output Layer $D$**. Similarly, we build a hypernetwork $\mathrm{hp}_{\mathrm{out}}\left(o_i[j]\right) = \mathrm{MLP}\left(o_i[j]\right)$ to generate the weight matrices $\mathcal{W}_{\mathrm{out}}$ for the output layer $D$. The $j$-th output of layer $D$ is computed as:

$$\pi_i(a_{\mathrm{equiv}}|o_i')[j] = h_i \mathrm{hp}_{\mathrm{out}}\left(o_i'[j]\right) \qquad o_i' = g o_i, \ \forall g \in G, \forall j \in \{1, \ldots, m\} \qquad (8)$$

By utilizing hypernetworks, the input $o_i[j]$ and the weight matrix $\mathcal{W}_{\mathrm{out}}[j]$ directly correspond one-to-one. The input order change will result in the same output order change, thus achieving PE. See Fig.4 for the whole architecture of HPN.

In this section, when describing our methods, we represent module $A$ and $D$ as networks with only one layer, but the idea of DPN and HPN can be easily extended to multiple layers. Besides, both DPN and HPN are general designs and can be easily integrated into existing MARL algorithms to boost their performance. All parameters of DPN and HPN are trained end-to-end with backpropagation according to the underlying RL loss function. Implementation details can be found in B.2.

# 5 EXPERIMENT

## 5.1 STARCRAFT MULTIAGENT CHALLENGE (SMAC AND SMACV2)

**Setup and codebase**. We first evaluate our methods in SMAC, which is an important testbed for MARL algorithms. SMAC consists of a set of StarCraft II micro battle scenarios, where units are divided into two teams: allies and enemies. The ally units are controlled by the agents while the enemy units are controlled by the built-in rule-based bots. The agents can observe the distance, relative location, health, shield and type of the units within the sight range. The goal is to train the agents to defeat the enemy units. We evaluate our methods in all Hard and Super Hard scenarios. Following Samvelyan et al. (2019); Hu et al. (2021a), the evaluation metric is a function that maps the environment steps to test winning rates. Each experiment is repeated using 5 independent training runs and the resulting plots show the median performance as well as the 25%-75% percentiles. Recently, Hu et al. (2021a) demonstrate that the optimized QMIX (Rashid et al., 2018) achieves the SOTA performance in SMAC. Thus, all codes and hyperparameters used in this paper are based on their released project PyMARL2. Detailed parameter settings are given in Appendix C. Except for SMAC, we also evaluate our HPN on a new challenging benchmark SMACv2. Due to the space limit, detailed settings and evaluation results are presented in Appendix A.10.

### 5.1.1 APPLYING HPN TO SOTA FINE-TUNED QMIX

We apply HPN to each $Q_i(a_i|o_i)$ of the fine-tuned QMIX and VDN (Hu et al., 2021a). The learning curves over 8 hard and super hard scenarios are shown in Fig.5. We conclude that: (1) HPN-QMIX surpasses the fine-tuned QMIX by a large margin and achieves 100% test win rates in almost all scenarios, especially in *5m_vs_6m, 3s5z_vs_3s6z* and *6h_vs_8z*, which has never been achieved before. Our HPN-QMIX achieves the new SOTA in SMAC; (2) HPN-VDN significantly improves the performance of fine-tuned VDN, and it even surpasses the fine-tuned QMIX in most scenarios, which minimizes the gaps between the VDN-based and QMIX-based algorithms; (3) HPN significantly improves the sample efficiency of both VDN and QMIX, especially in *5m_vs_6m, MMM2, 3s5z_vs_3s6z* and *6h_vs_8z*. In *5m_vs_6m*, to achieve the same 80% win rate, HPN-VDN and HPN-QMIX reduce the environmental interaction steps by a factor of $\frac{1}{4}$ compared with the counterparts.

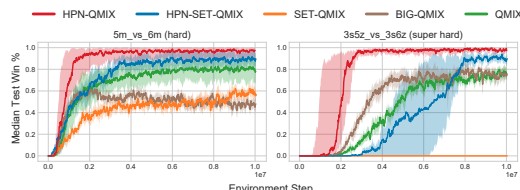

Figure 6: Comparisons of HPN and DPN against baselines considering the PI and PE properties.

### 5.1.2 COMPARISON WITH PI AND PE BASELINES

The baselines we compared include: (1) **DA-QMIX**: According to (Ye et al., 2020), we apply data augmentation to QMIX by generating more training data through shuffling the input order and using the generated data to additionally update the parameters. (2) **SET-QMIX**: we apply Deep Set (Li et al., 2021) to each $Q_i(a_i|o_i)$ of QMIX, i.e., all $x_i$s use a shared embedding layer and then aggregated by sum pooling, which can be considered as a special case of DPN, i.e., DPN(1). (3) **GNN-QMIX**: We apply GNN (Liu et al., 2020; Wang et al., 2020b) to each $Q_i(a_i|o_i)$ of QMIX to achieve PI. (4) **ASN-QMIX** (Wang et al., 2019): ASN-QMIX models the influences of different actions on other agents based on their semantics, e.g., move or attack, which is similar to the PE property considered in this paper. (5) **UPDeT-QMIX** (Hu et al., 2021b): a recently MARL algorithm based on Transformer which implicitly considers the PI and PE properties.

The results are shown in Fig.6. We conclude that: HPN > DPN ≥ UPDeT ≥ ASN > QMIX > GNN ≈ SET ≈ DA[6]. Specifically, (1) HPN-QMIX achieves the best win rates, which validates the effectiveness of our PI and PE design; (2) HPN-QMIX performs better than DPN-QMIX. The reason is that HPN-QMIX utilizes more flexible hypernetworks to achieve PI and PE with enhanced representational capacity. (3) UPDeT-QMIX and ASN-QMIX achieve comparable win rates with DPN-QMIX in most scenarios, which indicates that utilizing the PI and PE properties is important. (4) GNN-QMIX and SET-QMIX achieve similar performance. Although PI is achieved, GNN-QMIX and SET-QMIX still perform worse than vanilla QMIX, especially in *3s5z_vs_3s6z* and *6h_vs_8z*. This confirms that using a shared embedding layer $\phi(x_i)$ limits the representational capacity and restricts the final performance. (5) DA-QMIX improves the performance of QMIX in *3s5z_vs_3s6z* through much more times of parameter updating. However, its learning process is unstable and it collapses in all other scenarios due to the perturbation of the input features, which validates that it is hard to make a permutation-sensitive function (e.g., MLP) achieve PI through training solely. The learning curves of these PI/PE baselines equipped with VDN are shown in Appendix A.2, which have similar results. All implementation details are shown in Appendix B.2. Besides, in Appendix A.1, we also test and analyze these methods via a simple policy evaluation experiment.

### 5.1.3 ABLATION: ENLARGING THE NETWORK SIZE OF THE BASELINE

For HPN, incorporating hypernetworks leads to a 'bigger' model. To make a more fair comparison, we enlarge the network size of $Q_i(a_i|o_i)$ in fine-tuned QMIX (denoted as BIG-QMIX) such that it has more parameters than HPN-QMIX. Besides, the UPDeT baselines already have comparable or more parameters than HPN. The detailed sizes of these models are shown in Table 3 of Appendix A.3. From Fig.7, we see

Figure 7: Ablation studies.

that simply increasing the parameter number cannot improve the performance. In contrast, it even leads to worse results. VDN-based results are presented in Appendix A.3.

### 5.1.4 ABLATION: IMPORTANCE OF THE PE OUTPUT LAYER AND INPUT NETWORK CAPACITY

(1) To validate the importance of the PE output layer, we add the hypernetwork-based output layer of HPN to SET-QMIX (denoted as HPN-SET-QMIX). From Fig.7, we see that incorporating a PE output layer could significantly boost the performance of SET-QMIX. The converged performance of HPN-SET-QMIX is superior to that of vanilla QMIX. (2) However, due to the limited represen-

---

[6]We use the binary comparison operators here to indicate the performance order of these algorithms.

tational capacity of the shared embedding layer of Deep Set, the performance of HPN-SET-QMIX is still worse than HPN-QMIX. The only difference between HPN-SET-QMIX and HPN-QMIX is the number of embedding weight matrices in the input layer. The results validate that improving the representational capacity of the PI input layer is beneficial.

### 5.1.5 APPLYING HPN TO MAPPO AND QPLEX

The results of applying HPN to MAPPO (Yu et al., 2021) and QPLEX (Wang et al., 2020a) in 5m_vs_6m and 3s5z_vs_3s6z are shown in Fig.8. We see that HPN-MAPPO and HPN-QPLEX consistently improve the performance of MAPPO and QPLEX, which validates that HPN can be easily integrated into many types of MARL algorithms and boost their performance. Full results are shown in Appendix A.4.

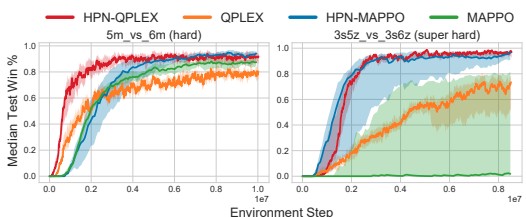

Figure 8: Apply HPN to QPLEX and MAPPO.

### 5.2 MULTIAGENT PARTICLE ENVIRONMENT (MPE)

We also evaluate the proposed DPN and HPN in the cooperative navigation and the predator-prey scenarios of MPE (Lowe et al., 2017), where the actions only consist of movements. Therefore, only the PI property is needed. We follow the experimental settings of PIC (Liu et al., 2020) (which utilizes GNN to achieve PI, i.e., GNN-MADDPG) and apply our DPN and HPN to the

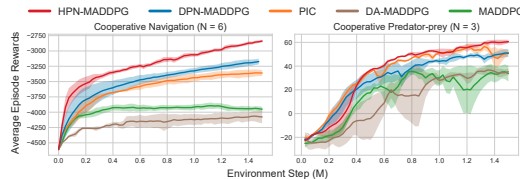

Figure 9: Apply HPN and DPN to MADDPG.

joint Q-function of MADDPG. We implement the code based on the official PIC codebase. The learning curves are shown in Fig.9. We see that our HPN-MADDPG outperforms the GNN-based PIC, DA-MADDPG and MADDPG baselines by a large margin, which validates the superiority of our PI designs. The detailed experimental settings and full results are presented in Appendix A.6.

### 5.3 GOOGLE RESEARCH FOOTBALL (GRF)

Finally, we evaluate HPN in two Google Research Football (GRF) (Kurach et al., 2020) academic scenarios: 3_vs_1_with_keeper and counterattack_hard. We control the left team players, which need to coordinate their positions to organize attacks, and only scoring leads to rewards. The observations consist of 5 parts: ball information, left team, right team, controlled player and match state. Each agent has 19 discrete ac-

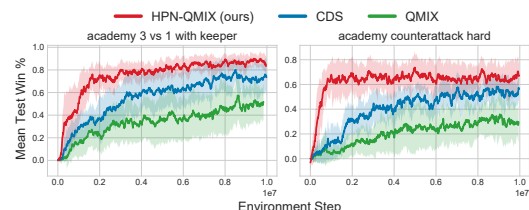

Figure 10: Apply HPN to GRF.

tions, including moving, sliding, shooting and passing, where the targets of the passing actions correspond to its teammates. Detailed settings can be found in Appendix A.7. We apply HPN to QMIX and compare it with the SOTA CDS-QMIX (Chenghao et al., 2021). We show the average win rate across 5 seeds in Fig.10. HPN can significantly boost the performance of QMIX and HPN-QMIX outperforms the SOTA method CDS-QMIX by a large margin in these two scenarios.

## 6 CONCLUSION

In this paper, we propose two PI and PE designs, both of which can be easily integrated into existing MARL algorithms to boost their performance. Although we only test the proposed methods in model-free MARL algorithms, they are general designs and have great potential to improve the performance of model-based (Yuan et al., 2022), multitask and transfer learning (Wu et al., 2023) algorithms. Besides, we currently follow the settings of (Qin et al., 2022; Wang et al., 2020b; Hu et al., 2021b; Long et al., 2019), where the configuration of the input-output relationships and the observation/state structure are set manually. For future works, it is interesting to automatically detect such structural information.

## ACKNOWLEDGMENTS

This work is supported by the National Natural Science Foundation of China (Grant No.62106172), the "New Generation of Artificial Intelligence" Major Project of Science & Technology 2030 (Grant No.2022ZD0116402), and the Science and Technology on Information Systems Engineering Laboratory (Grant No.WDZC20235250409, No.WDZC20205250407).

We thank Zipeng Dai for the insightful discussions and helpful writing suggestions. We are very grateful for the valuable and constructive comments of all reviewers.

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

# A   ADDITIONAL EXPERIMENTAL SETTINGS AND RESULTS

## A.1   A SIMPLE POLICY EVALUATION PROBLEM.

We present a simple policy evaluation experiment to show the influence of the model's representational capacity on the converged performance. We consider the following permutation-invariant (PI) models: ❶ Deep Set, ❷ DA-MLP (apply data augmentation to a permutation-sensitive MLP model), ❸ DPN (apply DPN to the same MLP model), ❹ HPN (apply HPN to the same MLP model) and ❺ Attention (use self-attention layers and a pooling function to achieve PI).

The experimental settings are as follows. There are 2 agents in total. Each agent $i$ only has one dimension of feature $x_i$. For the convenience of analyzing, we set that each $x_i$ is an integer and $x_i \in \{1, 2, ..., 30\}$, i.e., each agent $i$ only has 30 different features. Thus the size of the joint state space ($[x_1, x_2]$) after concatenating is $30 * 30 = 900$. To make the policy evaluation task permutation-invariant, we simply set the target value $Y$ of each state $[x_1, x_2]$ as $x_1 * x_2$.

In section 4.3, when introducing HPN, we asked a question that whether we can provide an infinite number of candidate weight matrices of DPN. In this simple task, we can directly construct a separate weight matrix for each feature $x_i$. Since $x_i$ is discrete (which is enumerable), we can explicitly maintain a parameter table and exactly record a different embedding weight for each different feature $x_i$. We denote this direct method as ❻ DPN($\infty$), which means 'DPN with infinite weights'. Our HPN uses a hypernetwok to approximately achieve 'infinite weight' by generating a different weight matrix for each different input $x_i$. We use the Mean Square Error (MSE) as the loss function to train these different models. The code for this simple experiment is also available at https://github.com/tjuHaoXiaotian/API-Network (see '*Synthetic_Policy_Evaluation.py*').

The comparison of the learning curves and the converged MSE losses of these different PI models are shown in Fig.11 and Table 1 respectively. We conclude that DPN($\infty$) > HPN > Attention > DPN > DeepSet > DA-MLP, where '>' means 'performs better than', which indicates that increasing the representational capacity of the model can help to achieve much less MSE loss.

Table 1: The comparison of the converged MSE losses of these different PI models.

|      | DA-MLP  | Deep Set | DPN    | Attention | HPN  | DPN($\infty$) |
| ---- | ------- | -------- | ------ | --------- | ---- | ------------- |
| MSE  | 1495.55 | 1401.23  | 119.54 | 49.12     | 6.34 | 0.37          |

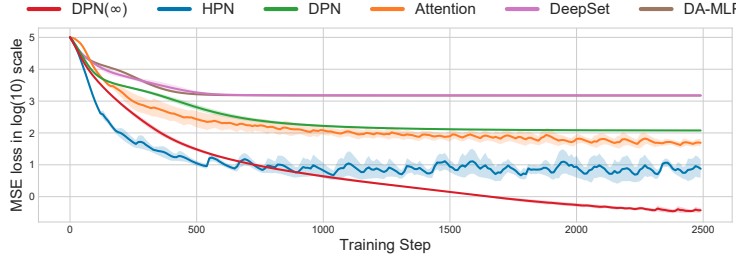

Figure 11: The comparison of the learning curves of different PI models.

If we keep each $x_i \in \{1, 2, ..., 30\}$ and simply increase the agent number, the changes of the original state space by simple concatenation and the reduced state space by using PI representations are shown in Table 2 below. We see that by using PI representations, the state space can be significantly reduced.

Table 2: Changes of the state space size with the increase of the agent number.

| agent number                 | 2        | 3          | 4            | 5             |
| ---------------------------- | -------- | ---------- | ------------ | ------------- |
| simple concatenation         | 900      | 27000      | 810000       | 24300000      |
| PI representation (percent)  | 465 (0.52) | 4960 (0.18) | 40920 (0.05) | 278256 (0.01) |

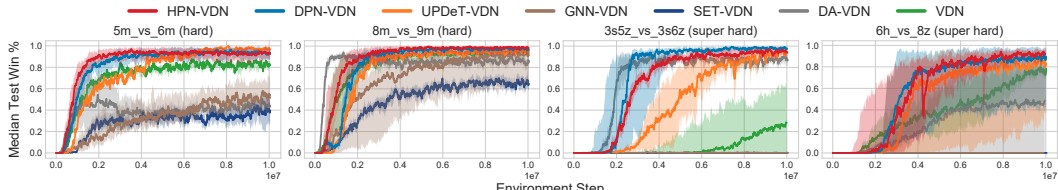

Figure 12: Comparisons of VDN-based methods considering the PI and PE properties.

## A.2 COMPARISON WITH VDN-BASED PI AND PE BASELINES

The learning curves of the PI/PE baselines equipped with VDN are shown in Fig.12, which demonstrate that: HPN ≥ DPN ≥ UPDeT > VDN > GNN ≈ SET ≈ DA[7]. Specifically, (1) HPN-VDN and DPN-VDN achieve the best win rates; (2) Since UPDeT uses a shared token embedding layer followed by multi-head self-attention layers to process all components of the input sets, the PI and PE properties are implicitly taken into consideration. The results of UPDeT-VDN also validate that incorporating PI and PE into the model design could reduce the observation space and improve the converged performance in most scenarios. (3) GNN-VDN achieves slightly better performance than SET-VDN. Although permutation-invariant is maintained, GNN-VDN and SET-VDN perform worse than vanilla QMIX, (especially in *3s5z_vs_3s6z* and *6h_vs_8z*, the win rates are approximate 0%). This confirms that the use of a shared embedding layer $\phi(x_i)$ for each component $x_i$ limits the representational capacities and restricts the final performance. (4) DA-VDN significantly improves the learning speed and performance of vanilla VDN in *3s5z_vs_3s6z* by data augmentation and much more times of parameter updating. However, the learning process is unstable, which collapses in all other scenarios due to the perturbation of the input features, which validates that it is hard to train a permutation-sensitive function (e.g., MLP) to output the same value when taking different orders of features as inputs.

## A.3 ABLATION STUDIES.

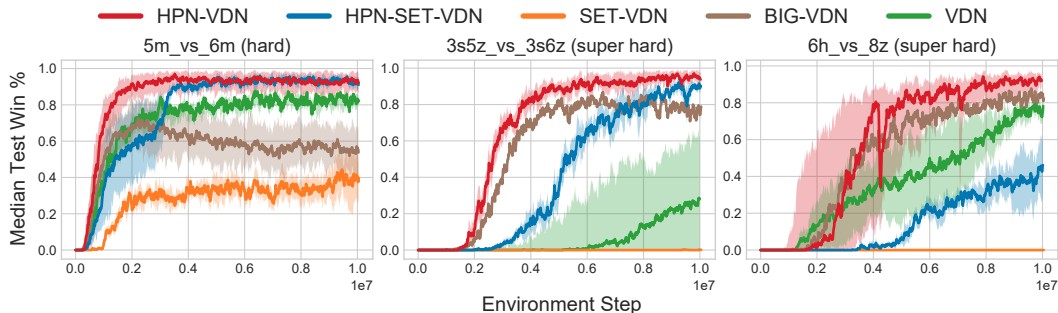

Figure 13: Ablation studies. All methods are equipped with VDN.

### A.3.1 ENLARGING THE NETWORK SIZE.

We also enlarge the agent network of vanilla VDN (denoted as BIG-VDN) such that the number of parameters is larger than our HPN-VDN. The detailed numbers of parameters are shown in Table 3. The results are shown in Fig.13. We see that simply increasing the parameter number cannot always guarantee better performance. For example, in 5m_vs_6m, the win rate of BIG-VDN is worse than the vanilla VDN. In 3s5z_vs_3s6z and 6h_vs_8z, BIG-VDN does achieve better performance, but the performance of BIG-VDN is still worse than our HPN-VDN in all scenarios.

### A.3.2 IMPORTANCE OF THE PE OUTPUT LAYER AND THE CAPACITY OF THE PI INPUT LAYER

To validate the importance of the permutation-equivariant output layer, we also add the hypernetwork-based output layer of HPN to SET-VDN (denoted as HPN-SET-VDN). The results are shown in Fig.13. We see that incorporating an APE output layer could significantly boost the

---

[7]We use the binary comparison operators here to indicate the performance order of these algorithms.

Table 3: The number of parameters of the individual Q-networks in VDN, QMIX, BIG-VDN, BIG-QMIX, HPN-VDN, HPN-QMIX, UPDeT-VDN and UPDeT-QMIX.

| Parameter Size | VDN(QMIX) | BIG-VDN(QMIX) | HPN-VDN(QMIX) | UPDeT-VDN(QMIX) |
|---|---|---|---|---|
| 5m_vs_6m | 30.412K | 109.964K | 72.647K | 96.294K |
| 3s_vs_5z | 29.707K | 108.555K | 81.031K | 96.246K |
| 8m_vs_9m | 32.911K | 114.959K | 72.839K | 96.438K |
| corridor | 39.262K | 127.646K | 76.999K | 96.39K |
| 3s5z_vs_3s6z | 36.175K | 121.487K | 98.375K | 96.582K |
| 6h_vs_8z | 32.206K | 113.55K | 76.935K | 96.342K |

performance of SET-VDN, and that the converged performance of HPN-SET-VDN is superior to the vanilla VDN in 5m_vs_6m and 3s5z_vs_3s6z.

However, due to the limited representational capacity of the shared embedding layer of Deep Set, the performance of HPN-SET-VDN is still worse than our HPN-VDN, especially in 6h_vs_8z. Note that the only difference between HPN-VDN and HPN-SET-VDN is the input layer, e.g., using hypernetwork-based customized embeddings or a simply shared one. The results validate the importance of improving the representational capacity of the permutation-invariant input layer.

## A.4 APPLYING HPN TO QPLEX AND MAPPO

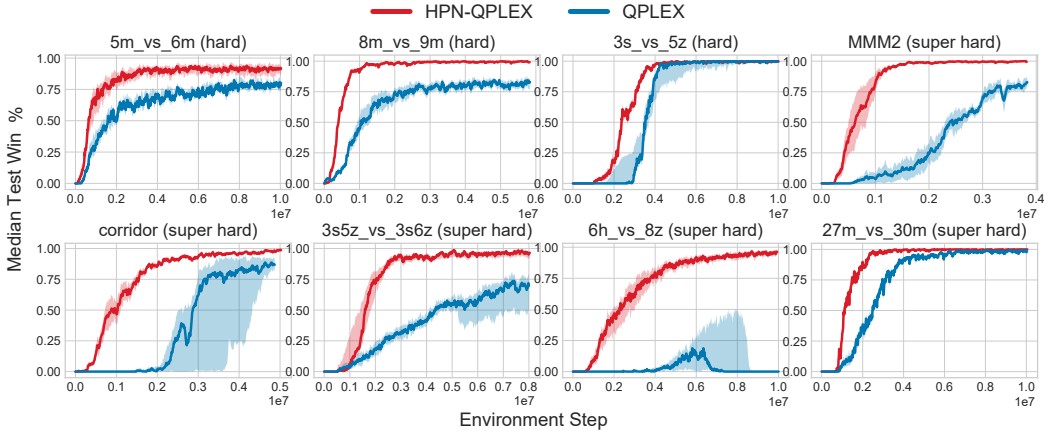

Figure 14: The learning curves of HPN-QPLEX compared with vanilla QPLEX in the hard and super hard scenarios of SMAC.

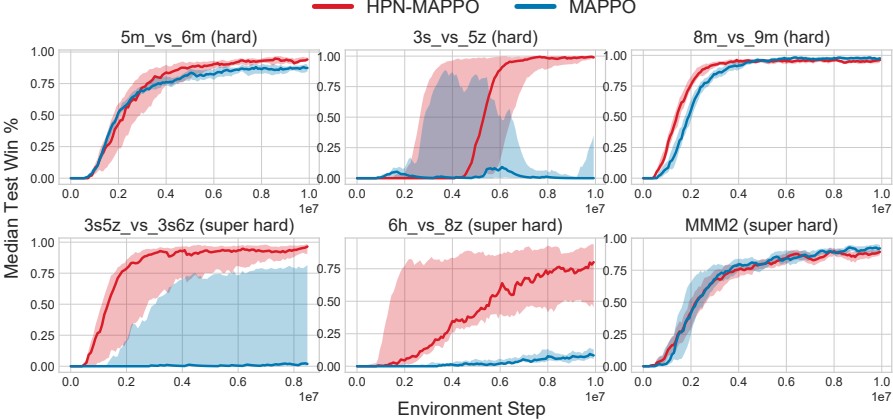

Figure 15: The learning curves of HPN-MAPPO compared with the vanilla MAPPO in the hard and super hard scenarios of SMAC.

To demonstrate that our methods can be easily integrated into many types of MARL algorithms and boost their performance, we also apply HPN to a typical credit-assignment method QPLEX (Wang et al., 2020a) (denoted as HPN-QPLEX) and a policy-based MARL algorithm MAPPO (Yu et al., 2021) (denoted as HPN-MAPPO). The results are shown in Fig.14 and Fig.15. We see that HPN significantly improves the performance of QPLEX and MAPPO, which validates that our method can be easily combined with existing MARL algorithms and improves their performance (especially for super hard scenarios).

## A.5 APPLYING HPN TO DEEP COORDINATION GRAPH.

Recently, Deep Coordination Graph (DCG) (Böhmer et al., 2020) scales traditional coordination graph based MARL methods to large state-action spaces, shows its ability to solve the relative over-generalization problem, and obtains competitive results on StarCraft II micromanagement tasks. Further, based on DCG, (Wang et al., 2021) proposes an improved version, named Context-Aware SparsE Coordination graphs (CASEC). CASEC learns a sparse and adaptive coordination graph (Wang et al., 2021), which can largely reduce the communication overhead and improve the performance. Besides, CASEC incorporates action representations into the utility and payoff functions to reduce the estimation errors and alleviate the learning instability issue.

Both DCG and CASEC inject the permutation invariance inductive bias into the design of the pairwise payoff function $q_{ij}(a_i, a_j|o_i, o_j)$. They achieve permutation invariance by permuting the input order of $[o_i, o_j]$ and taking the average of both. To show the generality of our method, we also apply HPN to the utility function and payoff function of CASEC and show the performance in Fig.16. The codes of CASEC and HPN-CASEC are also available at https://github.com/tjuHaoXiaotian/API-Network (see *code/src/config/algs/casec.yaml* and *code/src/config/algs/hpn_casec.yaml*).

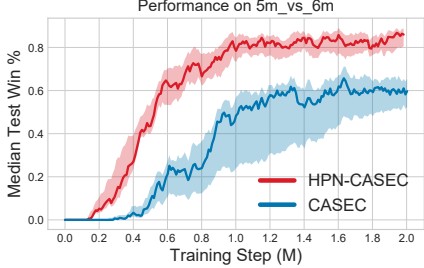

Figure 16: The learning curves of HPN-CASEC and CASEC in 5m_vs_6m.

In Fig.16, we compare HPN-CASEC with the vanilla CASEC in 5m_vs_6m. Results show that HPN can significantly improve the performance of CASEC, which validate that HPN is very easy to implement and can be easily integrated into many existing MARL approaches.

## A.6 MULTIAGENT PARTICLE ENVIRONMENT

We evaluate the proposed DPN and HPN on the classical Multiagent Particle Environment (MPE) (Lowe et al., 2017) tasks, where the actions only consist of movement actions. Therefore, only the permutation invariance property is needed. We follow the experimental settings of PIC (permutation-invariant Critic for MADDPG, which utilizes GNN to achieve PI, i.e., GNNMADDPG) (Liu et al., 2020) and apply our DPN and HPN to the centralized critic Q-function of MADDPG (Lowe et al., 2017). Each component $x_i$ represents the concatenation of agent i's observation and action. The input set $X_j$ contains all agents' observation-actions. We implement the code based on the official PIC. The baselines we considered are PIC (Liu et al., 2020), DA-MADDPG (Ye et al., 2020) and MADDPG (Lowe et al., 2017). The tasks we consider are as follows:

- **Cooperative navigation**: $n$ agents move cooperatively to cover L landmarks in the environment. The reward encourages the agents to get close to landmarks. An agent observes its location and velocity, and the relative location of the landmarks and other agents.
- **Cooperative predator-prey**: $n$ slower predators work together to chase M fast-moving prey. The predators get a positive reward when colliding with prey. Preys are environment

controlled. A predator observes its location and velocity, the relative location of the L landmarks and other predators and the relative location and velocity of the prey.

The learning curves of different methods in the cooperative navigation task (the agent number $n = 6$) and the cooperative predator-prey task (the agent number $n = 3$) are given in Fig.9. Besides, We further test HPN on two more cooperative navigation tasks with 100 and 200 agents respectively. The learning curves are shown in Figure 17. The results show that HPN-MADDPG can significantly improve the performance of vanilla MADDPG and achieves superior sample efficiency and converged performance than PIC. All experiments are repeated for five runs with different random seeds. We see that our HPN-MADDPG outperforms the PIC, DA-MADDPG and MADDPG baselines in these two tasks, which validates the superiority of our permutation-invariant designs.

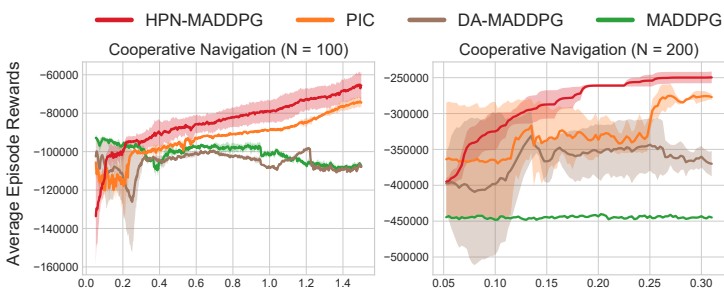

Figure 17: Comparisons of HPN-MADDPG against PIC, DA-MADDPG and MADDPG in cooperative navigation with 100 and 200 agents.

## A.7 GOOGLE RESEARCH FOOTBALL

We evaluate HPN in two Google Research Football (GRF) academic scenarios: 3_vs_1_with_keeper and counterattack_hard. In these tasks, we control the left team players except for the goalkeeper. The right team players are controlled by the built-in rule-based bots. The agents need to coordinate their positions to organize attacks and only scoring leads to rewards. The observations are factorizable and are composed of five parts: ball information, left team, right team, controlled player information and match state. Detailed feature lists are shown in Table 4. Each agent has 19 discrete actions, including moving, sliding, shooting and passing. Following the settings of CDS (Chenghao et al., 2021), we also make a reasonable change to the two half-court offensive scenarios: we will lose if our players or the ball returns to our half-court. All methods are tested with this modification. The final reward is +100 when our team wins, -1 when our player or the ball returns to our half-court, and 0 otherwise.

We apply HPN to QMIX and compare it with the SOTA CDS-QMIX (Chenghao et al., 2021). In detail, when applying HPN to QMIX, both the PI actions, e.g., moving, sliding and shooting, and the PE actions, e.g., long_pass, high_pass and short_pass are considered. For each player, since the targets of these passing actions directly correspond to its teammates, we apply the PE output layer to generate the Q-values of these passing actions, where the hypernetwork takes each ally player's features as input and generates the weight matrices for the passing actions. Besides, in the official GRF environment, as we cannot directly control which teammates the current player passes the ball to, we take a max pooling over all ally-related Q-values to get the final Q-values for the three passing actions. We show the average win rate across 5 seeds in Fig.10. HPN can significantly boost the performance of QMIX and our HPN-QMIX outperforms the SOTA method CDS by a large margin in these two scenarios.

## A.8 GENERALIZATION: CAN HPN GENERALIZE TO A NEW TASK WITH A DIFFERENT NUMBER OF AGENTS?

Apart from achieving PI and PE, another benefit of HPN is that it can naturally handle variable numbers of inputs and outputs. Therefore, as also stated in the conclusion section, HPN can be potentially used to design more efficient multitask learning and transfer learning algorithms. For example, we can directly transfer the learned HPN policy in one task to new tasks with different numbers of agents and improve the learning efficiency in the new tasks. Transfer learning results of

| Observation | Player | Absolute position |
| | | Absolute speed |
| | Left team | Relative position |
| | | Relative speed |
| | Right team | Relative position |
| | | Relative speed |
| | Ball | Absolute position |
| | | Belong to (team ID) |
| State | Left team | Absolute position |
| | | Absolute speed |
| | | Tired factor |
| | | Player type |
| | Right team | Absolute position |
| | | Absolute speed |
| | | Tired factor |
| | | Player type |
| | Ball | Absolute position |
| | | Absolute speed |
| | | Absolute rotate speed |
| | | Belong to (team ID) |
| | | Belong to (player ID) |

Table 4: The feature composition of the observation and the state in Google Research Football

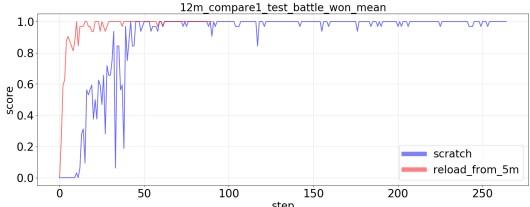

(a) Transfer learning results on 12m. Red: reload the learned policy in 5m to 12m and then continuously train the policy. Blue: learn from scratch.

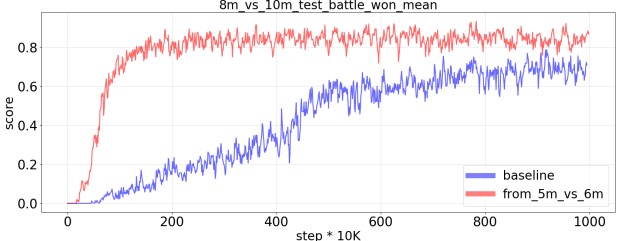

(b) Transfer learning results on 8m_vs_10m. Red: reload the learned policy in 5m_vs_6m to 8m_vs_10m and then continuously train the policy. Blue: learn from scratch.

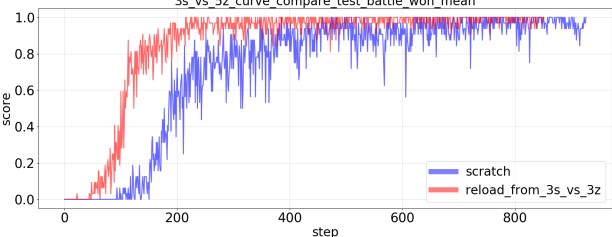

(c) Transfer learning results on 3s_vs_5z. Red: reload the learned policy in 3s_vs_3z to 3s_vs_5z and then continuously train the policy. Blue: learn from scratch.

Figure 18: Transferring the learned HPN-VDN policy in one task to a new task with a different number of agents.

5m → 12m, 5m_vs_6m → 8m_vs_10m, 3s_vs_3z → 3s_vs_5z are shown in Fig.18. We see that the previously trained HPN policies can serve as better initialization policies for new tasks.

## A.9 To achieve PI and PE, what if we just sort the entities according to distance from the focal agent?

(1) When we first started working on this project, we have also considered a similar baseline: we sort the entities according to (type, distance), i.e., according to their types first and then the relative distances if two entities' types are same. But we found that this solution do not always work well. Here, we provide the learning curves of HPN, QMIX, VDN, SORT-QMIX, and SORT-VDN in 4 hard and super hard scenarios on SMAC in Figure 19. The results show that the sorting baseline can slightly improve the performance of vanilla QMIX/VDN in 5m_vs_6m and 3s5z_vs_3s6z. However, in 8m_vs_9m and 6h_vs_8z, it harms the performance.

(2) The reason is that each entity has many types of features, e.g., relative x, relative y, relative distance, entity type, health point, shield, etc. Relative distance is just one of them. Simply sorting the entities by their relative distances while ignoring the influences of the other features may not be appropriate. Besides, as different x and y can have the same distance and different distances can have the same order, the same $o_i[j]$ may be arranged at different positions and be multiplied by different 'weight matrices' (according to $z_i = \sum_{j=1}^{m} o_i[j]\mathcal{W}_{\text{in}}[j]$). Therefore, learning may become unstable if we frequently reorder the inputs by distance only.

(3) Thus, our target is **not only matching the observation and action belonging to the same entity but stabilizing the learning process by always assigning the same weight matrix $\mathcal{W}_{\text{in}}[j]$, i.e., a stable weight, to the same entity features** $o_i[j]$ no matter where $o_i[j]$ is arranged. In this paper, we propose DPN and HPN to achieve this.

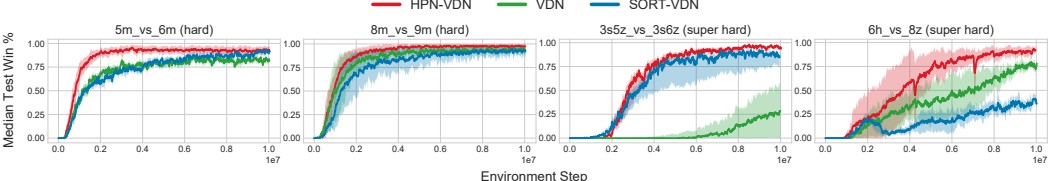

(a) Comparisons of HPN-VDN, VDN, and a baseline that sorts the entities in observation according to their distances from the focal agent (denoted as SORT-VDN).

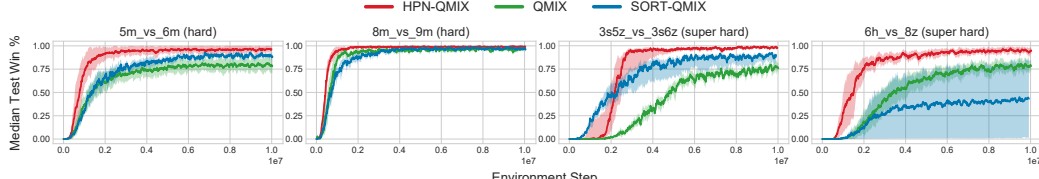

(b) Comparisons of HPN-QMIX, QMIX, and a baseline that sorts the entities in observation according to their distances from the focal agent (denoted as SORT-QMIX).

Figure 19: Comparisons of HPN with the sorting based baseline.

## A.10 Evaluate HPN on SMAC-v2

SMAC-v2 makes three major changes to SMAC: randomising start positions, randomising unit types, and restricting the agent field-of-view and shooting range to a cone. These first two changes increase more randomness to challenge contemporary MARL algorithms. The third change makes features harder to infer and adds the challenge that agents must actively gather information (require more efficient exploration). Since our target is not to design more efficient exploration algorithms, we keep the field-of-view and attack of the agents a full circle as in SMAC.

- **Random Start Positions**: Random start positions come in two different types. First, there is the *surrounded* type, where the allied units are spawned in the middle of the map, and

surrounded by enemy units. This challenges the allied units to overcome the enemies approach from multiple angles at once. Secondly, there are the *reflect_position* scenarios. These randomly select positions for the allied units, and then reflect their positions in the midpoint of the map to get the enemy spawn positions. Example figures are shown in Figure 20 below.

- **Random Unit Types**: Battles in SMACv2 do not always feature units of the same type each time, as they did in SMAC. Instead, units are spawned randomly according to certain pre-fixed probabilities. Units in StarCraft II are split up into different races. Units from different races cannot be on the same team. For each of the three races (Protoss, Terran, and Zerg), SMACv2 uses three unit types. Detailed generation probabilities are shown in Figure 21.

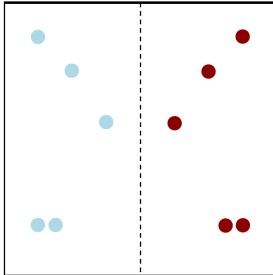 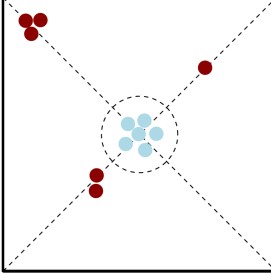

Figure 20: Examples of the two different types of start positions, opposite and surrounded. Allied units are shown in blue and enemy units in dark red.

| Race | Unit | Generation Probability |
|------|------|------------------------|
| Terran | Marine | 0.45 |
| | Marauder | 0.45 |
| | Medivac | 0.1 |
| Protoss | Stalker | 0.45 |
| | Zealot | 0.45 |
| | Colossus | 0.1 |
| Zerg | Zergling | 0.45 |
| | Hydralisk | 0.45 |
| | Baneling | 0.1 |

Figure 21: Detailed generation probabilities of the three types of units for the three races (Protoss, Terran, and Zerg).

**Our HPN can naturally handle the two types of new challenges.** Thanks to the PI and PE properties, our HPN is more robust to the randomly changed start positions of the entities. Thanks to the entity-wise modeling and using hypernetwork to generate a customized *weight matrix* for each type of unit, HPN can handle the randomly generated unit types as well. The comparisons of HPN-VDN with VDN on three difficult scenarios across the three races (Protoss, Terran, and Zerg) are shown in Figure 22. Results show that our HPN significantly improves the sample efficiency and the converged test win rates of the baseline VDN.

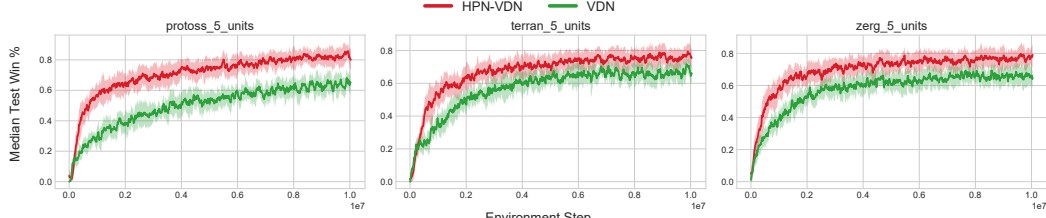

Figure 22: The learning curves of HPN-VDN and VDN in 3 difficult scenarios of SMAC-v2.

# B  TECHNICAL DETAILS

## B.1  DECENTRALIZED PARTIALLY OBSERVABLE MDP

We model a fully cooperative multiagent task as a Dec-POMDP (Oliehoek & Amato, 2016), which is defined as a tuple $\langle \mathcal{N}, \mathcal{S}, \mathcal{O}, \mathcal{A}, P, R, Z, \gamma \rangle$. $\mathcal{N}$ is a set of $n$ agents. $\mathcal{S}$ is the set of global states. $\mathcal{O} = \{\mathcal{O}_1, \ldots \mathcal{O}_n\}$ denotes the observation space for $n$ agents. $\mathcal{A} = \mathcal{A}_1 \times \ldots \times \mathcal{A}_n$ is the joint action space, where $\mathcal{A}_i$ is the set of actions that agent $i$ can take. At each step, each agent $i$ receives a private observation $o_i \in \mathcal{O}_i$ according to the observation function $Z(s, \mathbf{a}) : \mathcal{S} \times \mathcal{A} \rightarrow \mathcal{O}$, and produces an action $a_i \in \mathcal{A}_i$ by a policy $\pi_i(a_i|o_i)$. All agents' individual actions constitute a joint action $\mathbf{a} = \langle a_1, \ldots, a_n \rangle \in \mathcal{A}$. Then the joint action $\mathbf{a}$ is executed and the environment transits to the next state $s'$ according to the transition probability $P(s'|s, \mathbf{a})$. All agents receive a shared global reward according to the reward function $R(s, \mathbf{a})$. All individual policies constitute the joint policy $\boldsymbol{\pi} = \pi_1 \times \ldots \times \pi_n$. The target is to find an optimal joint policy $\boldsymbol{\pi}$ which could maximize the expected return $R_t = \sum_{t=0}^{T} \gamma^t r(s^t, \mathbf{a}^t)$, where $\gamma$ is a discount factor and $T$ is the time horizon. The joint action-value function is defined as $Q_{\boldsymbol{\pi}}(s_t, \mathbf{a}_t) = \mathbb{E}_{\boldsymbol{\pi}, P}[R_t|s_t, \mathbf{a}_t]$. Each agent's individual action-value function is denoted as $Q_i(o_i, a_i)$.

## B.2  IMPLEMENTATION DETAILS OF OUR APPROACH AND BASELINES

The key points of implementing the baselines and our methods are summarized here:

(1) **DPN and HPN**: The proposed two methods inherently support heterogeneous scenarios since the entity's 'type' information has been taken into each entity's features. And the sample efficiency can be further improved within homogeneous agents compared to fixedly-ordered representation. For MMM and MMM2, we implemented a permutation-equivariant 'rescue-action' module for the only Medivac agent, which uses similar prior knowledge to ASN and UPDeT, i.e., action semantics. To focus on the core idea of our methods, we omitted these details in the method section.

**The objective to train the weight selection network of PDN.** As stated in the last paragraph of Section 4, all parameters of DPN are trained end-to-end with backpropagation according to the RL loss function. The weight selection network and the other networks work cooperatively to minimize the overall RL loss function.

**How PI is achieved of DPN.** As described in Section 4.2, for each $o_i[j]$, the weight selection network outputs the probability of selecting each weight matrix. During the forward pass at training step $t$, given the parameter snapshot of the weight selection network, the output probability of selecting each weight matrix is fixed. For each $o_i[j]$, we select the weight matrix with the maximum probability. However, directly selecting the argmax index is not differentiable. To make the selection process trainable, we apply a Straight Through Estimator [7] to get the one-hot encoding of the argmax index. We denote it as $\hat{p}_{\text{in}}(o_i[j])$. The weight matrix with the maximum probability can be acquired by $\hat{p}_{\text{in}}(o_i[j]) \cdot \mathcal{W}_{\text{in}}$. As selecting the weight matrix with the maximum probability is a deterministic process, according to Equation (5), PI is guaranteed.

Besides, to encourage more exploration at the beginning of training, we also add small gumbel noises (Jang et al., 2016) to the 'logits' within the epsilon anneal time. Within this interval, PI cannot be strictly guaranteed. When the epsilon anneal schedule is over, PI will be strictly guaranteed. The Straight Through Estimator written in PyTorch is shown below:

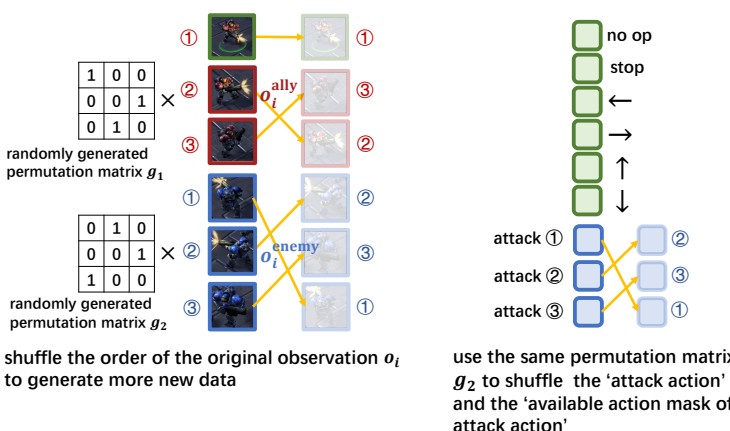

Figure 23: Applying Data Augmentation (DA) to the SMAC benchmark.

```
1  def straight_through(y_soft, dim):
2      # Straight Through Estimator.
3      index = y_soft.max(dim, keepdim=True)[1]
4      y_hard = th.zeros_like(y_soft).scatter_(dim, index, 1.0)
5      ret = y_hard - y_soft.detach() + y_soft
6      return ret
```

(2) **Data Augmentation (DA)** (Ye et al., 2020): we apply the core idea of Data Augmentation (Ye et al., 2020) to SMAC by randomly generating a number of permutation matrices to shuffle the 'observation', 'state', 'action' and 'available action mask' for each sample simultaneously to generate more training data. An illustration of the Data Augmentation process is shown in Fig.23. A noteworthy detail is that since the attack actions are permutation-equivariant to the enemies in the observation, the same permutation matrix $M_2$ that is utilized to permute $o_i^{\text{enemy}}$ should also be applied to permute the 'attack action' and 'available action mask of attack action' as well. The code is implemented based on PyMARL2 [8] for fair comparison.

(3) **Deep Set** (Zaheer et al., 2017; Li et al., 2021): the only difference between SET-QMIX and the vanilla QMIX is that the vanilla QMIX uses a fully connected layer to process the fixedly-ordered concatenation of the $m$ components in $o_i$ while SET-QMIX uses a shared embedding layer $h_i = \phi(x_i)$ to separately process each component $x_i$ in $o_i$ first, and then aggregates all $h_i$s by sum pooling. The code is also implemented based on PyMARL2 for fair comparison.

(4) **GNN**: Following PIC (Liu et al., 2020) and DyAN (Wang et al., 2020b), we apply GNN to the individual Q-network of QMIX (denoted as GNN-QMIX) to achieve permutation-invariant. The code is also implemented based on PyMARL2.

(5) **ASN** (Wang et al., 2019): we use the official code and adapt the code to PyMARL2 for fair comparison.

(6) **UPDeT** (Hu et al., 2021b): we use the official code [9] and adapt the code to PyMARL2 for fair comparison.

(7) **VDN and QMIX**: As mentioned in Section 3, vanilla VDN/QMIX uses fixedly-ordered entity-input and fixedly-ordered action-output (both are sorted by agent/enemy indices). Although VDN and QMIX do not explicitly consider the permutation invariance and permutation equivariance properties, they train a permutation-sensitive function to figure out the input-output relationships according to their fixed positions, which is implicit and inefficient.

The codes for the baselines are also published at https://github.com/tjuHaoXiaotian/API-Network.

---

[8]https://github.com/hijkzzz/pymarl2
[9]https://github.com/hhhusiyi-monash/UPDeT

## C  HYPERPARAMETER SETTINGS

For all MARL algorithms we use in SMAC (Samvelyan et al., 2019) (under the MIT License), we keep the hyperparameters the same as in PyMARL2 (Hu et al., 2021a) (under the Apache License v2.0). We list the detailed hyperparameter settings used in the paper below in Table 5 to help peers replicate our experiments more easily.

Table 5: Hyperparameter Settings of VDN-based or QMIX-based Methods.

| Parameter Name | Value |
|---|---|
| Exploration-related | |
| action_selector | epsilon_greedy |
| epsilon_start | 1.0 |
| epsilon_finish | 0.05 |
| epsilon_anneal_time | 100000 (500000 for *6h_vs_8z*) |
| Sampler-related | |
| runner | parallel |
| batch_size_run | 8 (4 for *3s5z_vs_3s6z*) |
| buffer_size | 5000 |
| t_max | 10050000 |
| Agent-related | |
| mac | hpn_mac for HPN, dpn_mac for DPN, set_mac for Deep Set, updet_mac for UPDeT and n_mac for others |
| agent | hpn_rnn for HPN, dpn_rnn for DPN, set_rnn for Deep Set, updet_rnn for UPDeT and rnn for others |
| HPN_hidden_dim | 64 (only for HPN) |
| HPN_layer_num | 2 (only for HPN) |
| permutation_net_dim | 64 (only for DPN) |
| Training-related | |
| softmax_tau | 0.5 (only for DPN) |
| learner | nq_learner |
| mixer | qmix or vdn |
| mixing_embed_dim | 32 (only for qmix-based) |
| hypernet_embed | 64 (only for qmix-based) |
| lr | 0.001 |
| td_lambda | 0.6 (0.3 for *6h_vs_8z*) |
| optimizer | adam |
| target_update_interval | 200 |

## D  COMPUTING ENVIRONMENT

We conducted our experiments on an Intel(R) Xeon(R) Platinum 8171M CPU @ 2.60GHz processor based system. The system consists of 2 processors, each with 26 cores running at 2.60GHz (52 cores in total) with 32KB of L1, 1024 KB of L2, 40MB of unified L3 cache, and 250 GB of memory. Besides, we use 2 GeForce GTX 1080 Ti GPUs to facilitate the training procedure. The operating system is Ubuntu 16.04.

## E  LIMITATIONS

We currently follow the settings of (Qin et al., 2022; Wang et al., 2020b; Hu et al., 2021b; Long et al., 2019; Wang et al., 2019), where the configuration of the input-output relationships and the observation/state structures are set manually.

**What if the observations are images or the structural information is not available?**

The high-level idea of this paper is to leverage some formats of symmetries to reduce the size of the search space. Since typical MARL benchmarks represent observations as factorizable vectors (which can provide more direct and compact information than images), we currently focus on the permutation symmetries, i.e., PI and PE.

For image inputs, the rotational or reflectional symmetries are more prominent characteristics. Thus, we could leverage rotation invariance or rotation equivariance to design better MARL algorithms, which is also a novel research direction.

For vector inputs, when the structural information is unknown, a potential solution is:

- (1) Learning action representations using a forward model. We want to learn action representations that can reflect the effects of actions on the environment and other agents. The effect of an action can be measured by the induced reward and the change in the states.

- (2) Using all actions' representations as queries and using all entities' embedded features (potentially generated by HPN) as keys and values, we leverage the self-attention mechanism to generate the Q-values of each action. Since the self-attention computation is invariant to the input entities' order, PI and PE are achieved. And the input-output relationships may be learned implicitly by the self-attention mechanism.

Automatically detecting such structural information is interesting and we leave this as future works.

