# OpenReview forum: "Boosting Multiagent Reinforcement Learning via Permutation Invariant and Permutation Equivariant Networks"
_ICLR.cc/2023/Conference — ICLR 2023 poster_

### Official Review · Reviewer_JeYo · 2022-10-21

**Confidence:** 4
**Correctness:** 3
**Technical Novelty And Significance:** 2
**Empirical Novelty And Significance:** 3
**Recommendation:** 6

**Clarity, Quality, Novelty And Reproducibility:**

**Clarity:** The paper is generally well-written and conveys the main insights well.
**Quality:** The amount of quality depends on how strong assumption #3 is.
**Novelty:** While PI and PE properties are separately studied in previous works, this work is the first paper that exploits the two properties simultaneously and explicitly.
**Reproducibility:** The source code is provided in the supplementary material to reproduce the results.

**Strength And Weaknesses:**

**Strengths:**
1. The proposed frameworks can be applied to existing MARL methods by following the minimal modification principle and modifying the input and output layers only.
2. Experiments are performed using the benchmark domains and show state-of-the-art performance compared to competitive baselines.
3. The paper is generally well-written and explained clearly.

**Weakness:**
This paper is built on three assumptions: 1) the observation and action space are factorized, 2) a different entity order in observation does not affect information (i.e., PI property), and 3) the agent has prior knowledge about the structures of its observation and action space (e.g., in Figure 1, the agent knows which information corresponds to ally and opponent agents and which actions correspond to PI and PE actions). While I agree with the authors that assumptions #1 and #2 generally hold in many MARL settings, assumption #3 can be a strong assumption and can limit the applicability of the proposed framework. For example, the deep set method exploits PI by requiring assumptions #1 and #2, which still hold in model-free MARL. However, to additionally exploit PE, this paper requires assumption #3, which is generally not available for model-free settings. If assumption #3 is strong, it would be beneficial to discuss which potential methods can be used to address assumption #3.

**Questions:**
1. The related work section states that the representation capacity of the deep set and GNNs is limited due to the use of the shared embedding layer. Do these methods have to use the shared embedding layer? If not, would it be easy to modify these baselines not to use the shared embedding layer and thus improve the representation capacity of these methods?
2. The problem statement is based on Dec-POMDP, the standard framework for collaborative settings. However, this paper uses competitive setting examples, such as SMAC, so I wonder why the problem statement is based on collaborative settings.
3. Regarding the PI Input Layer A in Section 4.2, the weight selection network is *learned* to select the corresponding weight matrix. Which objective is used to train the weight selection network? Also, because this network is learned, which can be noisy at the beginning of training, I would like to clarify how PI is guaranteed.

**Summary Of The Paper:**

This paper addresses the curse of the dimensionality problem in multiagent reinforcement learning (MARL), where the state-action space grows exponentially as the number of agents increases. To address this challenge, this paper proposes two novel implementations that exploit the permutation invariance (PI) and permutation equivariance (PE) properties. Specifically, the authors introduce the dynamic permutation network (DPN) and hyper policy network (HPN) that can guarantee PI and PE properties. Evaluations in SMAC, Google Research Football, and MPD show that the proposed frameworks outperform competitive baselines.

**Summary Of The Review:**

I would like to initially vote for 6 (marginally above the acceptance threshold). After the authors' response to my concern and questions, I will make a final decision on the recommendation.

---

> ### Author Response · Authors · 2022-11-19
> **Initial Response to Reviewer JeYo (Part 2/2)**
>
> ### 4. About how PI is achieved of DPN.
> **(1) During the forward pass at any training step $t$ (after the exploration phase), PI is strictly guaranteed.**
> * During the forward pass at training step $t$, given the parameter snapshot of the weight selection network, the output probability of selecting each weight matrix is fixed. For each $o_i[j]$, we select the weight matrix with the maximum probability. However, directly selecting the argmax index is not differentiable. To make the selection process trainable, we apply a Straight Through Estimator [7] to get the one-hot encoding of the argmax index. We denote it as $\hat{p}_\text{in}\left({o_i}[j]\right)$. The weight matrix with the maximum probability can be acquired by $\hat{p}_\text{in}\left({o_i}[j]\right) \cdot \mathcal{W}_\text{in}$. As selecting the weight matrix with the maximum probability is a deterministic process, according to Equation (5), PI is guaranteed.
> * At the beginning of training, to encourage more exploration, we add some small gumbel noises to the `logits` within the epsilon anneal time. Within this interval, PI cannot be strictly guaranteed. When the epsilon anneal schedule is over, PI will be strictly guaranteed according to (1). The Straight Through Estimator written in PyTorch is shown below.
> ```
> def straight_through(y_soft, dim):
>     index = y_soft.max(dim, keepdim=True)[1]
>     y_hard = th.zeros_like(y_soft).scatter_(dim, index, 1.0)
>     ret = y_hard - y_soft.detach() + y_soft
>     return ret
> ```
> **(2) During the backward pass at step $t$**, the parameters of the weight selection network is optimized to minimize the RL loss function.
>
> **(3) At step $t+1$**, although the parameters of the weight selection network are changed compared to that of step $t$, PI is still guaranteed given the new parameter snapshot. Each $o_i[j]$ will also be assigned with a fixed weight matrix (thus PI is ensured) but the weight matrix may be different from that of step $t$.
>
> We thank the reviewer for pointing out the unclear description of how PI is guaranteed in DPN. We have clarified this in Section 4.2 and Appendix B.2 of the revised version.
>
> ### 5. About the assumption #3.
> > 3) the agent has prior knowledge about the structures of its observation and action space
>
> As stated in the Conclusion and the Limitation Section of Appendix E, currently, we assume that the configuration of the input-output relationships is known a priori. Although this assumption may be a little strong for some realistic tasks, injecting suitable inductive bias into the algorithm design is an efficient way to improve performance. Previous works, e.g., UPDeT [8] and ASN [9], also use similar assumptions to improve the algorithms' performance on typical MARL benchmarks.
>
> We thank the reviewer's suggestion that it's better to discuss which potential methods can be used to address assumption #3. When the input-output relationships are not known beforehand, we may:
> * (1) Learning action representations using a forward model. We want to learn action representations that can reflect the effects of actions on the environment and other agents. The effect of an action can be measured by the induced reward and the change in the states.
> * (2) Using all actions' representations as queries and using all entities' embedded features (generated by HPN) as keys and values, we leverage the self-attention mechanism to generate the Q-values of each action. Since the self-attention computation is invariant to the input entities' order, PI and PE are achieved. And the input-output relationships may be learned implicitly by the self-attention mechanism.
>
> We have also added this discussion in the Limitation Section of Appendix E in our revised version. Automatically detecting such structural information is interesting and we leave this as future work.
>
> ---
>
> We hope our replies have addressed the concerns the reviewer posed and shown the improved quality of the paper. We are always willing to answer any of the reviewer's concerns about our work and we are looking forward to more inspiring discussions.
>
> ### Reference
> [1] Samvelyan M, et al. The StarCraft Multi-Agent Challenge. arXiv2019.
>
> [2] Rashid T, et al. QMIX: Monotonic Value Function Factorisation for Deep Multi-Agent Reinforcement Learning. ICML2018.
>
> [3] Böhmer W, et al. Deep Coordination Graphs. ICML2020.
>
> [4] Wang J, et al. Qplex: Duplex dueling multi-agent q-learning. ICLR2020.
>
> [5] Yu C, et al. The surprising effectiveness of ppo in cooperative multi-agent games. arXiv2021.
>
> [6] Hu J, et al. Riit: Rethinking the importance of implementation tricks in multi-agent reinforcement learning, arXiv2021.
>
> [7] Van Den Oord A, et al. Neural Discrete Representation Learning. NeurIPS2017.
>
> [8] Hu S, et al. Updet: Universal multi-agent reinforcement learning via policy decoupling with transformers. ICLR2020.
>
> [9] Wang W, et al. Action semantics network: Considering the effects of actions in multiagent systems. ICLR2019.

---

> > ### Comment · Reviewer_JeYo · 2022-11-19
> > **Response to Rebuttal**
> >
> > I appreciate the authors for their response. This response addresses my questions. I have a positive evaluation of this paper and maintain my score.

---

> > > ### Author Response · Authors · 2022-11-22
> > > **Many thanks to the reviewer**
> > >
> > > We are very grateful to the reviewer for the support and quick feedback on our work!

---

> ### Author Response · Authors · 2022-11-19
> **Initial Response to Reviewer JeYo (Part 1/2)**
>
> We thank the reviewer for the valuable and constructive comments. The questions proposed by the reviewer provide considerably helpful guidance to improve the quality of our paper.
>
> ### 1. Can we modify DeepSet or GNN to use unshared embedding layers?
> In fact, the PI input layers of DPN and HPN can be regarded as modified versions of DeepSet and GNN.
>
> If we denote $o_i=\left[x_1,\ldots x_m\right]^\mathsf{T}$, where $x_j$ is entity $j$'s features, the difference between DeepSet and HPN is:
>   * For DeepSet, each $x_j$ is mapped separately to a latent space using **a shared embedding layer** $\phi(x_j)$ and then merged by a PI pooling layer, e.g., $z_i=\Sigma_{j=1}^{m}\color{red}{\phi}(x_j)$.
>   * For HPN, different $x_j$s are processed by **different embedding layers**, e.g., $z_i=\Sigma_{j=1}^{m} \color{red}{\phi_j}(x_j)$, where the `weights` and `biases` parameters of $\phi_j$ are generated by a hypernetwork.
>   * Specifically, if $\phi(x_j)$ is a single layer network, in DeepSet, $z_i=\sum_{j=1}^m o_i[j] \cdot \color{red}{W_{\text {share}}}$. in HPN, $z_i=\sum_{j=1}^m o_i[j] \cdot \color{red}{W_{\text {in }}^{o_i[j]}}=\sum_{j=1}^m o_i[j] \cdot\color{red}{\mathrm{hp}_{\mathrm{in}}\left(o_i[j]\right)}$. Thus, the PI input layer of HPN can be considered as **Hyper-DeepSet (an improved version of DeepSet)**, where the unshared `weights` are generated by a hypernetwork.
>   * Experiments also validate the benefits of using Hyper-DeepSet. The ablation studies presented in Section 5.1.4 and Appendix A.3.2 show that the full HPN-QMIX > Hyper-DeepSet-QMIX (denoted as HPN-SET-QMIX) > DeepSet-QMIX (denoted as SET-QMIX). Note that the only difference between HPN-SET-QMIX and HPN-QMIX is whether using a shared $\color{red}{W_{\text {share}}}$ or unshared $\color{red}{W_{\text {in }}^{o_i[j]}}$ for each $o_i[j]$. The simple policy evaluation problem presented in Appendix A.1 also gives a detailed analysis about the benefits of using unshared $W_{\text {in }}^{o_i[j]}$.
>
> ### 2. SMAC can be considered as a fully collaborative game since we only control one side of the competitors.
> SMAC is naturally designed for benchmarking cooperative MARL algorithms [1]. In SMAC, the units are divided into two teams: allies and enemies. The enemy units are controlled by the built-in rule-based bots and thus can be considered as parts of the environment. The ally units are controlled by the learning agents whose target is to cooperatively defeat the enemy team. Therefore, from the side of the ally units, the game is fully cooperative. By this reason, SMAC has been used extensively by the MARL community for benchmarking cooperative MARL algorithms [1, 2, 3, 4, 5, 6].
>
>
> ### 3. The objective to train the weight selection network of PDN.
> As stated in the last paragraph of Section 4, all parameters of DPN are trained end-to-end with backpropagation according to the RL loss function. The weight selection network and the other networks work cooperatively to minimize the overall RL loss function.

---

### Official Review · Reviewer_TTsq · 2022-10-24

**Confidence:** 3
**Correctness:** 2
**Technical Novelty And Significance:** 3
**Empirical Novelty And Significance:** 2
**Recommendation:** 6

**Clarity, Quality, Novelty And Reproducibility:**

This paper is well-written and easy to follow. The proposed framework provides a novel and pluggable way to exploit the properties of PI and PE to reduce the dimensionality of state space in MARL.

**Strength And Weaknesses:**

**Strengths**

1. The problem this paper considers is important. How to handle the exponentially increasing dimensionality of state space in MARL is critical and this paper provides a new method to solve it.
2. The literature review is sufficient. Most previous works have been discussed and compared.
3. This framework is well-motivated. To solve the curse of dimensionality in state space, using the properties of permutation invariance and equivariance in MARL might be a natural way. Though previous works have discussed these properties, this work has an advantage over others that it only needs to modify the input and output layer.

**Weaknesses**

1. The proposed method is not sound. (1) As discussed in Sec.2, Deep Set & GNN have limited representation capacity, due to the use of shared embedding. However, the proposed framework also uses a shared embedding $h_i$. In Sec.4.3, the hyper net $\text{hp}_{\text{in}}$ and module B are all shared. (2) In Sec.4.2, the observation of an entity chooses a weight matrix via probability. There is no guarantee that the entity can always choose the same weight matrix because the probability will change when this entity is in a different state. (3) HyperNet is also shared. What is the difference between $h(o_i,\theta_h)^\top\cdot o_i$ and $f(o_i,\theta_f)$?
2. The empirical evaluation results can be improved. This work uses a complex method to match the observation and action belonging to the same entity. What if we just sort the entities according to distance from the focal agent? This might be a strong baseline.
3. The results are not significant enough compared with baselines considering PI and PE properties, such as UPDeT-QMIX, in Figure 6.
4. This paper does not provide any discussions of limitations. (1) Is there any observation space or action space that cannot be factored? (2) Do all actions in MARL belong to PI or PE? What if there is an action that can tie *two* enemies together with a rope? The entity-correlated action space might not be a general assumption.

**Summary Of The Paper:**

This paper focuses on breaking the curse of dimensionality in multi-agent state space by exploiting the permutation invariance and permutation equivariance inductive biases. To achieve this, they proposed two implementations: Dynamic Permutation Network and Hyper Policy Network. A key property of this framework is that they only need to modify the input/output layer, leaving the backbone module unchanged. This property makes the work easy to add to other works. Finally, extensive experiments in SMAC, Google Football and MPE show some strength of this framework.

**Summary Of The Review:**

This paper is interesting and the proposed method might be an easy way to be utilized to any other methods in the considered settings. However, the soundness and the empirical evaluations should be further verified.

---

> ### Author Response · Authors · 2022-11-19
> **Initial Response to Reviewer TTsq (Part 2/2)**
>
> ### 3. About DPN.
>
> > There is no guarantee that the entity can always choose the same weight matrix because the probability will change when this entity is in a different state.
>
> Our target is not to always assign the same `weight matrix` to the same entity. Instead, the target is to always assign the same `weight matrix` to the same features, i.e.,  $o_i[j]$. In DPN, the assignment probability for the same features $o_i[j]$ is fixed.
>
> ### 4. About the results of UPDeT.
> > The results are not significant enough compared with baselines considering PI and PE properties, such as UPDeT-QMIX, in Figure 6.
>
> In the following tables, we present the detailed win rates of HPN and UPDeT as the training goes on. Each element in the table represents the `median test win rate` at the corresponding step. From the tables, we see that the learning efficiency of HPN is significantly higher than the baseline UPDeT.
>
> The detailed win rates of QMIX-based algorithms (corresponding to Figure 6):
> | 5m_vs_6m | 1M | 2M | 3M | 4M | 5M |
> | -------- | -------- | -------- | -------- | -------- | -------- |
> |UPDeT-QMIX| 0.4      | 0.85     |0.90      | 0.91     | 0.91     |
> | HPN-QMIX | 0.74     | 0.92     |0.94      | 0.95     | 0.95     |
>
> | 3s5z_vs_3s6z | 1M | 2M | 3M | 4M | 5M |
> | -------- | -------- | -------- | -------- | -------- | -------- |
> |UPDeT-QMIX| 0.03     | 0.17     |0.51      | 0.70     | 0.83     |
> | HPN-QMIX | 0.01     | 0.49     |0.92      | 0.97     | 0.96     |
>
> | 6h_vs_8z | 1M | 2M | 3M | 4M | 5M |
> | -------- | -------- | -------- | -------- | -------- | -------- |
> |UPDeT-QMIX| 0        | 0.34     |0.66      | 0.82     | 0.89     |
> | HPN-QMIX | 0.31     | 0.77     |0.87      | 0.89     | 0.91     |
>
>
> The detailed win rates of VDN-based algorithms (corresponding to Figure 12 of the Appendix):
> | 5m_vs_6m | 1M | 2M | 3M | 4M | 5M |
> | ------- | -------- | -------- | -------- | -------- | -------- |
> |UPDeT-VDN| 0.22     | 0.61     |0.70      | 0.78     | 0.87     |
> | HPN-VDN | 0.69     | 0.90     |0.92      | 0.93     | 0.92     |
>
> | 3s5z_vs_3s6z | 1M | 2M | 3M | 4M | 5M |
> | ------- | -------- | -------- | -------- | -------- | -------- |
> |UPDeT-VDN| 0        | 0.01     |0.08      | 0.31     | 0.52     |
> | HPN-VDN | 0        | 0.09     |0.65      | 0.82     | 0.87     |
>
> | 6h_vs_8z | 1M | 2M | 3M | 4M | 5M |
> | -------  | -------- | -------- | -------- | -------- | -------- |
> |UPDeT-VDN | 0        | 0        |0.09      | 0.61     | 0.63     |
> | HPN-VDN  | 0        | 0.03     |0.40      | 0.68     | 0.80     |
>
>
>
> ### 5. About the limitations.
> We have discussed the limitations of the proposed methods both in the Conclusion Section and the Limitation Section of Appendix E. Currently, the configurations of the input-output relationships and the observation/state structures have to be set manually. For future works, it is interesting to automatically detect such structural information. In the revised version of Appendix E, we also add some discussions about how to handle the scenarios when the structural information and the input-output relationships are not available. Besides, we thank the reviewer for pointing out that there may exist actions in MARL that are neither PI nor PE. We have clarified the settings we considered in this paper in the revised version.
>
>
> ---
>
> We hope our replies and the updated experiments would be helpful in providing useful explanations and intuitive analysis. We are always willing to answer any of the reviewer's concerns and we sincerely thank the reviewer valuing the technical innovation and overall contributions of the paper. We are looking forward to more inspiring discussions.
>
>
>
> ### Reference
> [1] Rashid T, et al. QMIX: Monotonic Value Function Factorisation for Deep Multi-Agent Reinforcement Learning. ICML2018.
>
> [2] Christianos F, et al. Scaling Multi-Agent Reinforcement Learning with Selective Parameter Sharing. ICML2021.

---

> > ### Comment · Reviewer_TTsq · 2022-11-22
> > **Reply to the authors**
> >
> > Thanks for answering my questions. The explanation of the sorting baseline addressed my doubt. But I still have some doubts about other components.
> >
> > 1. Parameter sharing: Are the numbers of the parameters of hypernet and shared weight in baselines the same? The performance increase of the hypernet might be because of the increase in parameters.
> >
> > 2. The results of HPN-QMIX are incrementally higher than UPDeT-QMIX, but not significantly.

---

> > > ### Author Response · Authors · 2022-11-25
> > > **Thanks and further clarifications (part 2/2)**
> > >
> > > > Q2: The results of HPN-QMIX are incrementally higher than UPDeT-QMIX, but not significantly.
> > >
> > > **A2:**
> > > * **The learning efficiency of HPN-QMIX is much higher than UPDeT-QMIX.** As can be seen from the detailed win rates presented in our initial response (for convenience, we also summarize them in the following table), HPN-QMIX only needs about 3M steps to reach the results of UPDET-QMIX in 5M steps, i.e., the sample efficiency is improved by **40%**.
> > >
> > >   * The detailed win rates of QMIX-based algorithms (corresponding to Figure 6):
> > >     | 5m_vs_6m | 1M | 2M | 3M | 4M | 5M |
> > >     | -------- | -------- | -------- | -------- | -------- | -------- |
> > >     |UPDeT-QMIX| 0.4      | 0.85     |0.90      | 0.91     | 0.91     |
> > >     | HPN-QMIX | 0.74     | 0.92     |0.94      | 0.95     | 0.95     |
> > >
> > >     | 3s5z_vs_3s6z | 1M | 2M | 3M | 4M | 5M |
> > >     | -------- | -------- | -------- | -------- | -------- | -------- |
> > >     |UPDeT-QMIX| 0.03     | 0.17     |0.51      | 0.70     | 0.83     |
> > >     | HPN-QMIX | 0.01     | 0.49     |0.92      | 0.97     | 0.96     |
> > >
> > >     | 6h_vs_8z | 1M | 2M | 3M | 4M | 5M |
> > >     | -------- | -------- | -------- | -------- | -------- | -------- |
> > >     |UPDeT-QMIX| 0        | 0.34     |0.66      | 0.82     | 0.89     |
> > >     | HPN-QMIX | 0.31     | 0.77     |0.87      | 0.89     | 0.91     |
> > >
> > > * **UPDeT performs relatively worse in heterogeneous scenarios** (especially for tasks where the episode lengths required to win are much longer).
> > >   * For convenience, the figures of the detailed win rates and episode lengths of HPN and UPDeT in 5 hard scenarios of SMAC are shown in the following anonymous link: https://imgur.com/a/1DAXOeQ.
> > >   * From the figures, we see that UPDeT performs relatively worse in 3s5z_vs_3s6z and MMM2 where the scenarios are `heterogeneous` (i.e., there exists ≥2 types of units for each side) and `the episode length required to win is much longer than the others (i.e., 125 > 25)`. The agents have to handle heterogeneous entities and much longer input sequences.
> > >   * The advantages of HPN over UPDeT in such scenarios.
> > >     * For HPN, as we follow the Minimal Modification Principle, the backbone module is kept unchanged and a GRU is used to handle the long input sequences. Besides, as HPN uses hypernet to generate customized network parameters for heterogeneous entities, HPN can handle the different types of entities well.
> > >     * In contrast, UPDeT completely changes the model architecture and mainly focuses on how to handle the variable number of input entities (i.e., spatial info) via Transformer but pays relatively less attention to processing the temporal information. Therefore, the performance of UPDET degrades when the input sequence is long. Besides, as Transformer uses *shared weight matrices* to generate `queries, keys` and `values` for all input tokens, the performance is also compromised in such heterogeneous scenarios.
> > > * **Emphasizing the contributions of UPDeT and this paper.**
> > >   * **UPDeT is originally designed for transfer learning tasks**. UPDeT introduces a new transformer-based architecture for MARL that enables variable input and output sizes, which is used to train the agent on more diverse tasks for multitask training and is a great contribution to MARL.
> > >   * However, **UPDeT does not focus on `permutation invariance` or `permutation equivariance`.** PI and PE are not mentioned at all in the UPDeT paper. The reason why we consider UPDeT as a baseline is that the`Transformer` architecture can be utilized to achieve PI and PE and UPDeT is based on `Transformer`. Therefore, to the best of our knowledge, **we are the first to highlight the effectiveness of introducing both the PI and PE inductive biases on MARL**. This is also the contribution of this paper.
> > >   * Thus, we believe that both UPDeT and this paper make good contributions to the community of MARL from different aspects.
> > >
> > >
> > > We thank the reviewer for appreciating the technical innovation of appropriately injecting the PI and PE inductive biases into MARL following the minimal modification principle. Hope our responses can satisfactorily address your concerns. Thank you again for your time to provide us with valuable feedback.
> > >
> > >
> > >
> > > Best regards,
> > > Authors

---

> > > > ### Comment · Reviewer_TTsq · 2022-11-25
> > > > **Reply to the authors**
> > > >
> > > > Thanks for answering my questions. I’d like to raise the score.

---

> > > > > ### Author Response · Authors · 2022-11-25
> > > > > **Many thanks to the reviewer**
> > > > >
> > > > > Dear Reviewer TTsq,
> > > > >
> > > > > We are very grateful to the reviewer for the quick reply and the support for our work. Thank you!
> > > > >
> > > > > Best regards,
> > > > > Authors

---

> > > ### Author Response · Authors · 2022-11-25
> > > **Thanks and further clarifications (part 1/2)**
> > >
> > > Dear Reviewer TTsq,
> > >
> > > We appreciate the reviewer very much for timely feedback. Thank you for the valuable comments and suggestions on this paper, which make the paper better improved!
> > >
> > > We would like to address each of your concerns in the following.
> > >
> > > > Q1: Parameter sharing: Are the numbers of the parameters of hypernet and shared weight in baselines the same? The performance increase of the hypernet might be because of the increase in parameters.
> > >
> > > **A1:**
> > > * For HPN, as incorporating hypernetworks will lead to a ‘bigger’ network, in Section 5.1.3 and A.3.1 of the Appendix, we also provide an ablation study about enlarging the network size. We enlarge the shared individual Q-networks of VDN and QMIX (denoted as BIG-VDN and BIG-QMIX) such that they will have more parameters than HPN. Besides, the UPDeT-VDN and UPDeT-QMIX baselines compared in Figure 6 and Figure 12 already have comparable or more parameters than HPN.
> > > * The detailed number of parameters of these models are shown in Table 3 of Appendix A.3. For convenience, we summarize them in the following table. We see that BIG-VDN, BIG-QMIX, UPDeT-VDN, and UPDeT-QMIX have significantly more parameters than HPN-VDN and HPN-QMIX (except that HPN and UPDeT have a similar number of parameters in 3s5z_vs_3s6z.).
> > >
> > >
> > >     | Parameter Size of $Q_i(o_i)$ | VDN / QMIX | BIG-VDN / BIG-QMIX | HPN-VDN / HPN-QMIX |UPDeT-VDN / UPDeT-QMIX |
> > >     | --------   | -------- | -------- | -------- | -------- |
> > >     | 5m_vs_6m   |30.412K   |109.964K  |72.647K   |96.294K   |
> > >     | 8m_vs_9m   |32.911K   |114.959K  |72.839K   |96.438K   |
> > >     |3s5z_vs_3s6z|36.175K   |121.487K  |98.375K   |96.582K   |
> > >     | 6h_vs_8z   |32.206K   |113.55K   |76.935K   |96.342K   |
> > >   > Note that as the individual Q-networks of VDN-based and QMIX-based methods (e.g., UPDeT-VDN and UPDeT-QMIX) are the same, they share the same parameter size in the above table.
> > > * After analyzing the above table and the learning results shown in Figures 6, 7, 12, and 13, we conclude that simply increasing the parameter number cannot guarantee better performance. For example, in 5m vs 6m, the win rates of BIG-VDN and BIG-QMIX are lower than the vanilla VDN and QMIX. HPN-VDN and HPN-QMIX significantly outperform BIG-VDN and BIG-QMIX by large margins. Besides, although UPDeT has more parameters than HPN, HPN achieves better sample efficiency and converged performance than UPDeT. Some recent works [1, 2, 3, 4] also found that increasing the number of parameters cannot fundamentally improve the performance.
> > >
> > > ---
> > >
> > > ### Reference
> > > [1] Tonghan Wang, et al. ROMA: Multi-Agent Reinforcement Learning with Emergent Roles. ICML-2020.
> > >
> > > [2] Jiahan Cao, Lei Yuan, et al. LINDA: Multi-Agent Local Information Decomposition for Awareness of Teammates. arXiv2021.
> > >
> > > [3] Jianhao Wang, et al. Towards understanding cooperative multi-agent q-learning with value factorization. NeurIPS-2021.
> > >
> > > [4] Lei Yuan, Jianhao Wang, et al. Multi-Agent Incentive Communication via Decentralized Teammate Modeling. AAAI-2022.

---

> ### Author Response · Authors · 2022-11-19
> **Initial Response to Reviewer TTsq (Part 1/2)**
>
> We thank the reviewer for the valuable and constructive comments. The questions proposed by the reviewer provide considerably helpful guidance to improve the quality of our paper.
>
> ### 1. About parameter sharing.
> > Deep Set & GNN have limited representation capacity, due to the use of shared embedding. However, the proposed framework also uses a shared embedding $h_i$. In Sec.4.3, the hyper net $\text{hp}_\text{in}$ and module B are all shared.
>
> (1) We do not use shared embedding layers for each $x_j$ in observation $o_i=\left[x_1,\ldots x_m\right]^\mathsf{T}$, where $x_j$ is entity $j$'s features. The difference between DeepSet and DPN/HPN is:
>   * For DeepSet, each $x_j$ is mapped separately to a latent space using **a shared embedding layer** $\phi(x_j)$ and then merged by a PI pooling layer (e.g. sum) to ensure the PI of the whole function, e.g., $z_i=\Sigma_{j=1}^{m}\color{red}{\phi}(x_j)$.
>   * For DPN/HPN, different $x_j$s are  processed by **different embedding layers**, e.g., $z_i=\Sigma_{j=1}^{m} \color{red}{\phi_j}(x_j)$, where the `weights` and `biases` parameters of $\phi_j$ are either selected from a finite parameter set (in DPN) or generated by a hypernetwork (in HPN).
>   * Specifically, when $\phi(x_j)$ is a single layer network, in DeepSet, $z_i=\sum_{j=1}^m o_i[j] \cdot \color{red}{W_{\text {share}}}$. in HPN, $z_i=\sum_{j=1}^m o_i[j] \cdot \color{red}{W_{\text {in }}^{o_i[j]}}=\sum_{j=1}^m o_i[j] \cdot\color{red}{\mathrm{hp}_{\mathrm{in}}\left(o_i[j]\right)}$. Thus, the PI input layer of HPN can be considered as **Hyper-DeepSet**, where the unshared `weights` are generated by a hypernetwork.
>   * The ablation studies presented in Section 5.1.4 and Appendix A.3.2 show that the HPN-QMIX > Hyper-DeepSet-QMIX (HPN-SET-QMIX) > DeepSet-QMIX (SET-QMIX). Note that the only difference between HPN-SET-QMIX and HPN-QMIX is whether using a shared $\color{red}{W_{\text {share}}}$ or unshared $\color{red}{W_{\text {in }}^{o_i[j]}}$ for each $o_i[j]$. Besides, the simple policy evaluation experiment presented in Appendix A.1 also gives more detailed analysis about the benefits of using unshared $W_{\text {in }}^{o_i[j]}$.
>
> (2) For shared $h_i$ and module B, as shown in Figure 2/3/4, $h_i$ indicates the output of module B, and module A, B, C and D together form the policy network of agent $i$. In MARL, **sharing the policy network's parameters among agents** can significantly speed up the learning, which is a common practice [1, 2]. Here we follow such a common setting and share the policy network among agents for all methods used in this paper. However, this setting is not necessary for our method and we can use unshared policy networks for different agents.
>
> ### 2. About the sorting baseline.
> > This work uses a complex method to match the observation and action belonging to the same entity. What if we just sort the entities according to distance from the focal agent? This might be a strong baseline.
>
> (1) In the original environment, the entities and actions are sorted according to their indexes, thus the features and action belonging to the same entity are also matched. But this original observation space size is $\left|\mathcal{X}\right|^{m}$.
>
> (2) To reduce the observation space, when we first started working on this project, we have also considered a similar baseline: we sort the entities according to (type, distance), i.e., according to their types first and then the relative distances if two entities' types are same. But we found that this solution does not always work well. We have updated the results in Appendix A.9 of our revised version. For continuity of reading, here we present the learning curves of HPN, QMIX, VDN, SORT-QMIX, and SORT-VDN in 4 hard and super hard scenarios on SMAC in an anonymous link: [https://imgur.com/a/hckRHry](https://imgur.com/a/hckRHry). The results show that the sorting baseline can slightly improve the performance of vanilla QMIX/VDN in 5m_vs_6m and 3s5z_vs_3s6z. However, in 8m_vs_9m and 6h_vs_8z, it harms the performance.
>
> (3) The reason is that each entity has many types of features, e.g., relative x, relative y, relative distance, entity type, health point, shield, etc. Relative distance is just one of them. Simply sorting the entities by their relative distances while ignoring the influences of the other features may not be appropriate. Besides, as different x and y can have the same distance and different distances can have the same order, the same $o_i[j]$ may be arranged at different positions and be multiplied by different `weight matrices`. Therefore, frequently reordering the inputs by distance will result in the learning process becoming unstable.
>
> (4) Our method can **not only match the observation and action belonging to the same entity but stabilize the learning process by always assigning the same weight matrix $\mathcal{W}_{\text {in }}[j]$, i.e., a stable weight, to the same entity features $o_i[j]$** no matter where $o_i[j]$ is arranged.

---

### Official Review · Reviewer_RFWq · 2022-10-25

**Confidence:** 4
**Correctness:** 3
**Technical Novelty And Significance:** 3
**Empirical Novelty And Significance:** 3
**Recommendation:** 6

**Clarity, Quality, Novelty And Reproducibility:**

### Clarity:
The paper is well-written and easy to follow.

### Quality:
Overall, the paper is technically sound, but empirical results on settings with more agents are needed to support the claims about scalability.

### Novelty:
Using hyper-nets to achieve PI and PE is novel and interesting.

### Reproducibility:
The code is provided and the hyper-parameters are detailed in the appendix. The reviewer thinks there is a high chance one could reproduce the results.



**Strength And Weaknesses:**

### Strength:

1. While PI and PE for RL and MARL are widely studied, the idea of using hyper-net to output weight for different input and thus achieve PI and PE are new and interesting.

2. The reviewer appreciates that the proposed approach could be applied to existing architecture without changing the backbone. This is an advantage over existing graph-based PE and PI MARL approaches.

3. It is impressive that the proposed approach outperforms baselines on many super-hard SMAC tasks.

4. The paper is well-written and easy to follow.


### Weakness / Questions:

1. The reviewer has concerns on the assumption that the structural information of observations and the actions’ PI / PE property are available.  This is privileged information that is not used by baselines such as QMIX. This information may not be available in many real-world applications. For instance, suppose the observations are images or some feature vectors that are not interpretable. How do we apply the proposed approach?

2. The reviewer has some concern regarding the scalability. Specifically, suppose there are 100 agents, does the hyper-net need to generate 100 different weights?

3. While the paper highlights ‘break the curse of dimensionality’ and scalability as main contributions, the experiments only report results with few agents. For instance, in MPE, only six agents are considered. In contrast, baseline approaches such as PIC could actually scale to hundreds of agents. Results with more agents are needed to support the claims.


**Summary Of The Paper:**

This paper studies permutation invariance (PI) and permutation equivariance (PE) for multi-agent reinforcement learning. Different from existing works which achieve PI and PE via shared embedding of graph nets, the authors proposed a hypernet-based architecture, which is referred to as Hyper Policy Network (HPN). With HPN, the authors built PI input and PE output modules which could be plugged into existing MARL architectures without modifying the backbone network. The proposed approach is evaluated on SMAC, Google Research Football and MPE. The results show that the proposed approach outperforms baseline in various tasks.


**Summary Of The Review:**

In summary, the reviewer appreciates the proposed HPN, which nicely leverages the structural information of observation and builds the helpful inductive bias to achieve better performance. However, the reviewer thinks the limitations, such as assumption of the structural information of observation, should be more carefully discussed. In addition, empirical results on tasks with more agents are needed to support the claims.

---

> ### Author Response · Authors · 2022-11-19
> **Initial Response to Reviewer RFWq (Part 2/2)**
>
> ### 3. About the scalability of HPN.
> (1) HPN is computationally efficient with the help of GPUs. If we denote $o_i=\left[x_1,\ldots x_m\right]^\mathsf{T}$, where $x_j$ indicates entity $j$'s features in the observation, in our practical implementation, HPN treats $o_i$ as a batch of entities' features and generate their `weights` in a batch mode. With the help of advanced GPUs, this computation process is also efficient.
>
> (2) When the entity number $m$ in the observation is extremely high, e.g.,k $m=100$, we could borrow the idea of modifying the MPE environment in PIC [6]. The modified version of MPE in PIC could scale to hundreds of agents. They reduce the observation dimension by restricting that each agent can only observe the nearest $K$ neighboring entities, i.e., set $m=K$. With similar ideas, the computation efficiency of HPN can also be significantly improved.
>
> ### 4. Experiments on MPE with 100 and 200 agents.
> (1) In this paper, the experiment with the largest number involves 27 learning agents (the 27m_vs_30m scenario in SMAC), which is at a medium-scale. In this scenario, HPN scales well.
>
> (2) According to the reviewer's suggestion, we further test HPN on two more cooperative navigation tasks with 100 and 200 agents respectively. We have updated the new results in Appendix A.6 of the revised version. Here, for convenience, we also present the learning curves in an anonymous link: [https://imgur.com/a/jZc5ykG](https://imgur.com/a/jZc5ykG). The results show that  HPN-MADDPG can significantly improve the performance of vanilla MADDPG and achieves superior sample efficiency and converged performance than PIC.
>
>
> ---
>
> We hope our replies and the updated experiments would help address the concerns of the reviewer. We are always willing to answer any of the reviewer's questions and are looking forward to the following inspiring discussions.
>
>
> ### Reference
> [1] Qin R, et al. Multi-agent policy transfer via task relationship modeling. arXiv2022.
>
> [2] Hu S, et al. Updet: Universal multi-agent reinforcement learning via policy decoupling with transformers. ICLR2020.
>
> [3] Wang W, et al. Action semantics network: Considering the effects of actions in multiagent systems. ICLR2019.
>
> [4] Wang W, et al. From few to more: Large-scale dynamic multiagent curriculum learning. AAAI2020.
>
> [5] Long Q, et al. Evolutionary population curriculum for scaling multi-agent reinforcement learning. ICLR2019.
>
> [6] Liu I J, et al. PIC: permutation invariant critic for multiagent deep reinforcement learning. CoRL2019.

---

> ### Author Response · Authors · 2022-11-19
> **Initial Response to Reviewer RFWq (Part 1/2)**
>
> We thank the reviewer for the valuable and constructive comments, which provide considerably helpful guidance to improve the quality of our paper.
>
> ### 1. About the assumption that the structural information is available.
>
> **The assumption is reasonable for typical multi-agent benchmarks.** As stated in the Conclusion and the Limitation (section E of the Appendix) parts, currently, this paper focuses on how to properly integrate the PI and PE inductive biases into existing MARL algorithms, based on the assumption that the input-output relationships and the observation/state structure are known a priori. Although this assumption may be a little strong for some realistic tasks, injecting suitable inductive bias into the algorithm design is an efficient way to improve performance. Previous works, e.g., MATTAR [1], UPDeT[2], ASN[3], DyAN[4], and EPC[5], also use similar assumptions to improve the algorithms' performance on typical MARL benchmarks, e.g., SMAC, MPE, MAgent and Neural MMO, which is reasonable.
>
> We thank the reviewer for the question. In the following, we try to discuss how to handle the scenarios when the structural information is not available.
>
> ### 2. Suppose the observations are images or the structural information is not available.
>
> The high-level idea of this paper is to leverage some formats of symmetries to reduce the size of the search space. Since typical MARL benchmarks represent observations as factorizable vectors (which can provide more direct and compact information than images), in this paper, we focus on the permutation symmetries, i.e., PI and PE.
>
> **For image inputs**, the rotational or reflectional symmetries are more prominent characteristics. Thus, we could leverage rotation invariance or rotation equivariance to design better MARL algorithms, which is also a novel research direction.
>
> **For vector inputs**, when the input-output relationships are unknown, a potential solution is:
> * (1) Learning action representations using a forward model. We want to learn action representations that can reflect the effects of actions on the environment and other agents. The effect of an action can be measured by the induced reward and the change in the states.
> * (2) Using all actions' representations as queries and using all entities' embedded features (generated by HPN) as keys and values, we leverage the self-attention mechanism to generate the Q-values of each action. Since the self-attention computation is invariant to the input entities' order, PI and PE are achieved. And the input-output relationships may be learned implicitly by the self-attention mechanism.
>
> We have added this discussion in the Limitation Section of Appendix E in our revised version. Automatically detecting such structural information is interesting and we leave this as future work.

---

### Official Review · Reviewer_ZC3P · 2022-10-25

**Confidence:** 3
**Correctness:** 4
**Technical Novelty And Significance:** 3
**Empirical Novelty And Significance:** 3
**Recommendation:** 8

**Clarity, Quality, Novelty And Reproducibility:**

The paper is very clearly written in my opinion. The source code is provided. The experiments are very high quality. The novelty is not tremendously high, but I think it is a high impact use case.

**Strength And Weaknesses:**

Strengths:
- This paper is well aligned with the conference by demonstrating the value of improved representation learning in complex MARL domains.
- The PI and PE inductive biases are indeed very common characteristics of MARL problems, making correctly encoding them a fundamental research direction.
- The paper is clearly written and the proposed approach is clearly justified theoretically.
- The experiments are very well done. The domains considered are large-scale and commonly used, relevant baselines are considered, important ablations are considered, and the approach is shown to be largely agnostic of the choice of MARL backbone.

Weaknesses:
- The DPN and HPN architectures are not tremendously novel and utilize well known concepts from the representation learning literature.

**Summary Of The Paper:**

This paper looks at the intersection of representation learning and MARL. The authors aim to design network architectures that are permutation invariant to reordering of agents in the input representation layer of a MARL network and also consider how to develop some output units that are only equivariant to permutations. The authors propose two architectures towards this end (DPN and HPN) with the idea of keeping the backbone intact, allowing for reuse across a variety of different MARL algorithms. The authors demonstrate significant gains in sample efficiency using HPN and DPN across popular MARL environments and algorithms including on SMAC, MPE, and GRF.

**Summary Of The Review:**

As a reviewer, I found this paper to be a delight. It is clearly written, clearly motivated, and offers a focused yet potentially high impact solution. The experiments are also of extremely high caliber, really providing me with exactly what I would look for as a reviewer.  Ultimately, my consideration was more between strong accept and accept than it was considering that this paper should not be accepted. I believe it will make a nice contribution to the conference and believe it is likely to have a high impact on this subject area. My only concern, which is why I went with accept, is that the novelty of the proposed approach is really not that high in light of past work considering these topics in representation learning. However, I definitely think that this paper makes a strong contribution to the MARL community.

Update After Author Feedback:

I agree with the characterization provided by the authors of the source of novelty in their contribution and believe it is a nice addition to the community.

---

> ### Author Response · Authors · 2022-11-19
> **Initial Response to Reviewer ZC3P**
>
> Many thanks for the reviewer’s strong support of our work! We are very glad to see that our carefully prepared work can be recognized by the reviewer. Thanks for the valuable and uplifting comments!
>
> ### 1. About the Novelty of DPN and HPN.
> We agree with the reviewer that both the two architectures utilize existing techniques (e.g., hypernet) to achieve PI and PE, which are not tremendously novel. Our target is not to design a universal representation learning method. The idea of appropriately injecting the PI and PE inductive biases into MARL while following the minimal modification principle is novel, which makes the proposed methods become plug-and-play modules for existing MARL algorithms. Besides, using hypernet to achieve PI and PE is quite new and well performed.
>
> Thank the reviewer again for valuing the technical innovation and overall contributions of the paper. Hope that our work could have a high impact on the MARL community.

---

### Author Response · Authors · 2022-11-19
**General Response**

**We appreciate all the reviewers’ constructive suggestions and insightful comments.** Individual responses and a revised version of our paper have been uploaded. Modifications in the revised version are marked in red.

For the revision, major updates are summarized below:

* According to reviewer RFWq's suggestion, we add **two more experiments on MPE with 100 and 200 agents**. The results are presented in Appendix A.6. HPN-MADDPG achieves the best performance on these 2 cooperative navigation tasks.
* According to reviewer TTsq's advice, in Appendix A.9, we **add two sorting-based baselines, i.e., SORT-VDN and SORT-QMIX**, in the SMAC experiments. We sort the entities in each agent's observation according to the (type, distance), i.e., according to their types first and then the relative distances if the two entities' types are the same. Results show that sorting the entities does not always work well and sometimes even harms the performance.
* According to reviewer JeYo's advice, we add **more implementation details of DPN** in Appendix B.2.
* Currently, this paper focuses on how to properly integrate the PI and PE inductive biases into existing MARL algorithms, based on the assumption that the input-output relationships and the observation/state structure are known a priori. According to reviewer RFWq and TTsq's advice, in Appendix E, we **add some discussions about how to handle the scenarios when the structural information and the input-output relationships are not available**.
* Finally, we **further evaluate our HPN on a new challenging benchmark [SMAC-v2](https://github.com/oxwhirl/smacv2) [1].** SMAC-v2 adds more randomness to SMAC, e.g., randomizing start positions and randomizing unit types, which increases the difficulty of the game. Thanks to the PI and PE properties, our HPN is more robust to the randomly changed start positions of the entities. Thanks to the entity-wise modeling and using hypernetwork to generate a customized `weight matrix` for each type of unit, HPN can handle the randomly generated unit types as well. Results show that our HPN significantly improves the performance of the baseline method. We provide the detailed results in Appendix A.10. For continuity of reading, we also provide the learning curves in an anonymous link here: [https://imgur.com/a/iE9HS4r](https://imgur.com/a/iE9HS4r).


We hope our replies have addressed the questions and concerns the reviewers posed and shown the improved quality of the paper. **We are always willing to answer any of the reviewers' questions** and we sincerely hope the reviewers to value the technical innovation and overall contributions of the paper. We are looking forward to following inspiring discussions.


[1] SMACv2: A New Benchmark for Cooperative Multi-Agent Reinforcement Learning. GitHub repository: https://github.com/oxwhirl/smacv2

---

### Decision · Program_Chairs · 2023-01-20

**Decision:**

Accept: poster

**Justification For Why Not Higher Score:**

The method is somewhat of a straightforward application of results known in other areas of machine learning, in particular from the representation learning community.

**Justification For Why Not Lower Score:**

Solid experimental evaluation, including new results that were added during the reviewer period.


**Metareview: Summary, Strengths And Weaknesses:**

This paper addresses permutation invariance and equivariance in Dec-POMDPs. In a nutshell, as the number of agents increases there is a large number of identical states due to the arbitrary ordering of the agents.
The authors suggest two methods, Dynamic Permutation Network (DPN) and a Hyper Policy Network (HPN), to address the issue and show convincing performance improvements.

Strength:
Solid experimental evaluation, including new results that were added during the reviewer period.

Weakness:
 The method is somewhat of a straightforward application of results known in other areas of machine learning, in particular from the representation learning community.



**Note From Pc:**

if the above contains the word "oral" or "spotlight" please see: "oral" presentation means -> notable-top-5% and "spotlight" means -> notable-top-25%. As stated in our emails, we are disassociating presentation type from AC recommendations